# Human eIF2A has a minimal role in translation initiation and in uORF-mediated translational control in HeLa cells

**Mykola Roiuk[1,2,3], Marilena Neff[1,2,3], Aurelio A Teleman[1,2,3]***

[1]German Cancer Research Center (DKFZ) Heidelberg, Heidelberg, Germany; [2]Faculty of Medicine, Heidelberg University, Heidelberg, Germany; [3]Faculty of Biosciences, Heidelberg University, Heidelberg, Germany

## eLife Assessment

In this **valuable** study, Roiuk et al combined ribosome profiling and reporter assays to provide **compelling** evidence that eIF2A does not have a major impact on mRNA translation in HeLa cells. These findings are consistent with several recent publications that disaffirm the previously proposed role of eIF2A in directing protein synthesis under stress. Considering that stress-dependent perturbations in translation play a major role in homeostasis and several pathological states (e.g., cancer and neurological disorders), this work should be of broad interest to researchers studying regulation of gene expression, stress-adaptation, cancer and neurobiology.

**\*For correspondence:**
a.teleman@dkfz.de

**Competing interest:** The authors declare that no competing interests exist.

**Abstract** Translation initiation in eukaryotes requires a 40 S ribosome loaded with initiator tRNA which scans for an initiation codon. The initiator tRNA is usually recruited to the ribosome as part of a ternary complex composed of initiator tRNA, eIF2, and GTP. Although initiator tRNA recruitment was originally ascribed to another factor, eIF2A, it was later disproven and shown to occur via eIF2. Nonetheless, eIF2A is still considered a translation initiation factor because it binds the ribosome and shows genetic interactions with other initiation factors such as eIF4E. The exact function of eIF2A during translation initiation, however, remains unclear. Here, we use ribosome profiling and luciferase reporter assays to systematically test in HeLa cells the role of eIF2A in translation initiation, including translation of upstream ORFs. Since eIF2A is thought to take over the function of eIF2 when eIF2 is inhibited, we also test conditions where the integrated stress response is activated. In none of our assays, however, could we detect a role of eIF2A in translation initiation. It is possible that eIF2A plays a role in translation regulation in specific conditions that we have not tested here, or that it plays a role in a different aspect of RNA biology.

## Introduction

For cells and organisms to grow and develop properly, they require a machinery that translates mRNAs in a finely tuned manner. Alterations in the amount or activity of components of the translational machinery lead to severe pathological conditions (*Tahmasebi et al., 2018*). Of all the steps of protein production, translation initiation is considered to be the most precisely regulated and rate-limiting (*Palmiter, 1975*). Translation initiation under normal physiological conditions is well studied and occurs through a cap-dependent mechanism (*Hinnebusch and Lorsch, 2012*) that relies on the recruitment of the 40 S ribosomal subunit preloaded with a set of initiation factors to the 5'-end of

mRNAs. Correct positioning of the 40 S at the 5'-end enables scanning, i.e., movement of the 40 S in a 5' to 3' direction in search of the first initiation codon (AUG) within an optimal context (*Pestova and Kolupaeva, 2002*). The ribosome uses the CAU anticodon of the initiator Methionine tRNA (tRNA$_i^{Met}$) to identify the AUG start codon (*Kozak, 1991*). In the most common initiation pathway, tRNA$_i^{Met}$ is delivered to the 40 S via the ternary complex which consists of eIF2 bound to GTP and the tRNA$_i^{Met}$ itself. However, several alternative initiation factors have been reported to possess the ability to also interact and potentially deliver initiator tRNA: eIF2D, the DENR/MCTS1 complex, and eIF2A (reviewed in *Grove et al., 2024*). We focus here on eIF2A, since its function is still elusive.

eIF2A was initially considered to be a functional analog of prokaryotic IF2 (*Shafritz and Anderson, 1970*), however, later this role was reassigned to the above-mentioned heterotrimeric factor eIF2 (α, β, γ) (*Levin et al., 1973*). Unlike eIF2, eIF2A was proposed to recruit tRNA$_i^{Met}$ to the 40 S in a GTP-independent but codon-dependent manner (*Zoll et al., 2002*), however, this activity was later challenged (*Dmitriev et al., 2010*) by attributing it to eIF2D which co-purified with eIF2A in the initial study. Unlike eIF2, eIF2A is dispensable for organismal viability, as single knock-outs of *eIF2A* in model organisms such as *Saccharomyces cerevisiae* (*Komar et al., 2005*), *C. elegans* (*Kim et al., 2018*) and *M. musculus* (*Anderson et al., 2021*; *Golovko et al., 2016*) show little or no phenotype. However, double knock-outs of *eIF2A* and *eIF5B* exhibit severe growth defects both in yeast and in *C. elegans* (*Kim et al., 2018*; *Zoll et al., 2002*) and eIF2A is synthetic lethal with eIF4E in yeast (*Komar et al., 2005*). These genetic interactions suggest a role for eIF2A in translation initiation, however, the precise function of eIF2A is still not well defined, with contradictory results being reported. For example, eIF2A has been studied in the context of internal ribosome entry sites (IRES), where it was proposed to act both as a suppressor and an activator of IRES-mediated initiation. In yeast, eIF2A was reported to inhibit translation of URE2, GIC1, and PAB1 IRESes (*Reineke et al., 2008*; *Reineke and Merrick, 2009*). In Huh7 cells, knock-down of eIF2A was reported to hamper translation of a Hepatitis C virus (HCV) IRES reporter upon ER-stress (*Kim et al., 2011*), while a subsequent study detected no effect on the HCV-IRES reporter (*Jaafar et al., 2016*). Similarly, the knock-out of eIF2A in HAP1 cells did not affect translation of either a HCV-IRES reporter or a EMCV-IRES reporter (*González-Almela et al., 2018*), nor did it affect the infection rate of Sindbis virus (*Sanz et al., 2019*). Beyond IRES-mediated translation, eIF2A was reported to promote initiation from near-cognate start sites, initiated at leucine codons (CUG, GUG), by delivering elongator leucine tRNA$^{Leu}$ to the 40 S subunit (*Sendoel et al., 2017*; *Starck et al., 2012*; *Starck et al., 2016*). Such eIF2A-driven non-AUG initiation events were proposed to play a crucial role in different aspects of cell physiology and disease progression: cellular adaptation during the integrated stress response (*Chen et al., 2019*; *Starck et al., 2016*), fine-tuning of mitochondrial function (*Liang et al., 2014*), tumor progression (*Sendoel et al., 2017*), and repeat-associated non-AUG translation in familial amyotrophic lateral sclerosis (*Sonobe et al., 2018*). Such non-AUG initiation by eIF2A implies that eIF2A should possess high affinity for leucine tRNAs, however, according to in vitro filter binding assays, eIF2A binds inefficiently to elongator tRNA$^{Leu}$ compared to initiator tRNA$^{Met}$ (*Kim et al., 2018*), although, as mentioned above, it is questionable whether eIF2A has any tRNA binding capacity at all. Furthermore, loss of *eIF2A* in several systems did not recapitulate these effects on non-AUG initiation in either non-stressed or stress conditions (caused either by amino acid depletion or sodium arsenate treatment) (*Gaikwad et al., 2024*; *Ichihara et al., 2021*). In particular, one study *Gaikwad et al., 2024* found that eIF2A has little or no role in translation initiation and uORF mediated-translation in yeast. Since in some cases mRNA translation in human cells can differ from mRNA translation in yeast, whether eIF2A also has such a minimal role in human cells remains to be clarified.

Due to these various discrepancies, we decided to study the role of eIF2A on mRNA translation and cellular fitness in human cells. For this, we generated *eIF2A* knock-out HeLa cells and applied ribosome profiling to analyze changes in mRNA translation. Overall, we find little contribution of eIF2A to cellular translation both under normal and stressed conditions in HeLa cells.

# Results

## eIF2A-KO HeLa cell lines have no proliferative defect or change in global translation rates

To study the impact of eIF2A on cellular translation, we used CRISPR/Cas9 to generate two independent eIF2A knockout (*eIF2A*$^{KO}$) HeLa cell lines, in which different exons of *eIF2A* were targeted. We confirmed the absence of eIF2A by immunoblotting (*Figure 1A*) and by genotyping the clones (*Figure 1—figure supplement 1A*). eIF2A$^{KO}$ lines have reduced levels of *eIF2A* mRNA, likely arising from nonsense-mediated decay, due to premature stop codons in the *eIF2A* mRNA (*Figure 1—figure supplement 1B*). The eIF2A$^{KO}$ cells exhibit no defect in cellular proliferation rates (*Figure 1B*), which agrees with eIF2A having a dispensable role in organismal viability (*Anderson et al., 2021*; *Golovko et al., 2016*; *Kim et al., 2018*; *Komar et al., 2005*). To test if the loss of *eIF2A* affects global translation rates, we performed an *O*-propargyl-puromycin (OPP)-incorporation assay, which labels newly synthesized peptides and allows subsequent detection with anti-puromycin antibody. OPP incorporation, however, did not show any significant change in *eIF2A*$^{KO}$ cells compared to wild-type isogenic controls (*Figure 1C–D*). In line with the OPP-incorporation assay, we did not observe changes in polysome profiles of *eIF2A*$^{KO}$ cells compared to controls (*Figure 1E–F*).

Several studies have reported that stress can upregulate eIF2A levels (*Panzhinskiy et al., 2021*; *Starck et al., 2016*), or cause eIF2A to shuttle from the nucleus into the cytosol (*Kim et al., 2011*). Interestingly, *Grove et al., 2023* recently reported that increased levels of eIF2A in an in vitro translation system can suppress global translation initiation by directly binding and sequestering 40 S ribosomal subunits. Thus, an increase in cytosolic eIF2A, either due to increased total levels or due to shuttling out of the nucleus, could be a potential mechanism to suppress translation upon stress. To determine if either localization or global levels of eIF2A change in response to stress in HeLa cells, we performed subcellular fractionation and loaded on a gel nuclear and cytosolic fractions in proportion to their abundance in a cell (i.e. both fractions were lysed in equal volumes). In HeLa cells, eIF2A is present in both the cytosol and the nucleus, with higher levels in the cytosol (*Figure 1—figure supplement 1C*). Nonetheless, stress caused by tunicamycin treatment did not affect the subcellular distribution of eIF2A (*Figure 1—figure supplement 1C–D*), nor overall eIF2A levels (*Figure 1—figure supplement 1F–G*). To assess the localization of eIF2A with an orthogonal approach, we performed cell immunostaining. Since the eIF2A antibodies we have are not suitable for immunostaining, we overexpressed N-terminally FLAG-tagged eIF2A in HeLa cells treated cells with either DMSO or 100 μM sodium arsenite (SA) for 1 hr (condition used in *Kim et al., 2011*). This revealed that overexpressed eIF2A was excluded from the nucleus, and showed a cytosolic, speckled staining pattern, which was not affected by SA treatment (*Figure 1—figure supplement 1E*). These results are consistent with what has been observed for endogenous eIF2A in HAP1 cells (*González-Almela et al., 2018*; *Sanz et al., 2017*). To check if cellular stresses affects eIF2A levels in HeLa cells, we treated cells with different agents at concentrations and timeframes that were previously reported to alter eIF2A levels (*Starck et al., 2016*). However, we did not record significant changes in eIF2A levels with any of these treatments (*Figure 1—figure supplement 1F–G*). Nonetheless, we decided to test whether elevated levels of eIF2A would have the capacity to inhibit global translation levels in vivo. To this end, we overexpressed eIF2A and quantified global translation rates by OPP incorporation but did not observe any significant change (*Figure 1—figure supplement 1I-H*). In sum, our results indicate that loss of eIF2A as well as eIF2A ectopic overexpression in HeLa cells has little or no impact on global translation and cellular proliferation.

## Ribosome profiling identifies no eIF2A-dependent transcripts

We next investigated whether loss of *eIF2A* causes a change in translation of specific mRNAs which might be overlooked when assaying total global translation. To do this, we performed ribosome profiling, which sequences the mRNA footprints protected by ribosomes, on control and *eIF2A*$^{KO}$ cells. Normalization of the footprint counts for each mRNA relative to the abundance of that mRNA in total RNA yields an estimate of a transcript's translation efficiency. We used *eIF2A*$^{KO}$ clone 1, since the exon targeted in this clone is further 5' than in clone 2, thereby yielding a short, truncated protein of only 24 amino acids (*Figure 1—figure supplement 1A*). Correlation analysis revealed that both footprint counts and total RNA counts were highly reproducible across technical triplicates in both control and *eIF2A*$^{KO}$ cells (*Figure 2—figure supplement 1A*). In addition, our data revealed the expected

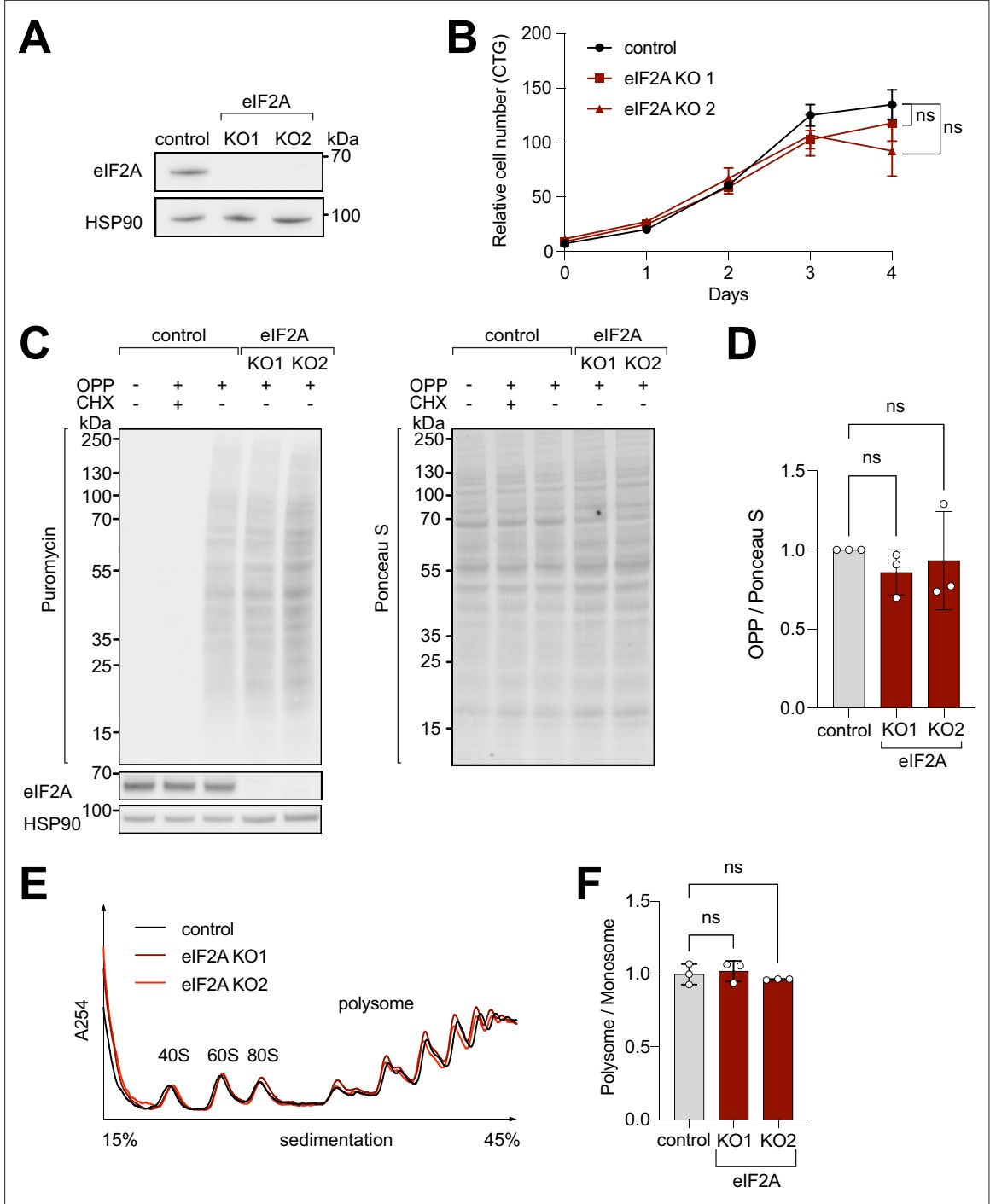

**Figure 1.** eIF2A has minimum effect on cell proliferation and global translation. (**A**) Validation that *eIF2A* knockout cells have no eIF2A protein by immunoblotting. (**B**) Two independent eIF2A knockout HeLa cell lines have no proliferation defect, assayed by CellTiter Glo. Error bars: standard deviation. Significance by ordinary one-way ANOVA. (**C–D**) eIF2A knockout HeLa cells have no detectable change in global translation rates compared to control cells. (**C**) The translation rate was measured by immunoblotting to detect O-propargyl-puromycin (OPP) incorporated by metabolic labeling (left), and normalized to total protein amount assayed by Ponceau S (right). Three independent replicates are quantified in panel (**D**). CHX = cycloheximide treated sample to completely block global translation (negative control). Error bars: standard deviation. Significance by ordinary one-way ANOVA. (**E–F**) Polysome profiles of *eIF2A*^KO cells show little to no difference to profiles from control cells. Lysates from either control or eIF2A^KO HeLa cells were separated on a sucrose gradient. One representative graph is shown in panel E. The polysome/80 S ratio of three independent replicates is shown in panel F. Error bars: standard deviation. Significance by ordinary one-way ANOVA.

The online version of this article includes the following source data and figure supplement(s) for figure 1:

*Figure 1 continued on next page*

*Figure 1 continued*

**Source data 1.** pdf: Uncropped western blots.

**Source data 2.** Original immunoblot files from Chemidoc.

**Figure supplement 1.** Loss of eIF2A does not perturb cell proliferation and global translation.

**Figure supplement 1—source data 1.** pdf: Uncropped western blots.

**Figure supplement 1—source data 2.** pdf: Uncropped western blots.

**Figure supplement 1—source data 3.** Original immunoblot files from Chemidoc.

enrichment of footprints in the coding sequence versus UTRs, as well as distinct triplet periodicity, both hallmarks of ribosome profiling data (*Figure 2—figure supplement 1B and C*). Metagene analysis showed similar global profiles in footprint densities within ORFs in wild-type and *eIF2A*$^{KO}$ samples, in agreement with the data presented above that eIF2A is dispensable for global mRNA translation (*Figure 2—figure supplement 1B,C*). A per-gene analysis identified only 15 genes in addition to eIF2A whose translation efficiency changes in *eIF2A*$^{KO}$ cells compared to control cells (*Figure 2A*). This is very different from our previous footprinting studies where we found that loss of DENR leads to reduced translation of 517 mRNAs (*Bohlen et al., 2020*), loss of PRRC2A/B/C leads to altered translation of 109 mRNAs (*Bohlen et al., 2023*), and expression of 4E-BP leads to altered translation of 605 mRNAs (*Roiuk et al., 2024*), suggesting that in comparison eIF2A plays a minor role in translational regulation (*Figure 2A*). We decided to validate some of these eIF2A-dependent candidates by immunoblotting, and to our surprise, amongst all the proteins we tested, only CCND3 showed reduced levels in one of the two *eIF2A*$^{KO}$ clones (*Figure 2B and C*). We then quantified mRNA levels for all the genes that we tested (*Figure 2D*). For NCAPH2, PPFIA1, and RPS6KB2, mRNA levels were either unchanged or reduced in the *eIF2A*$^{KO}$ cells, indicating that translation efficiency for these genes was either unchanged or increased upon loss of eIF2A, which does not agree with the ribosome profiling data. For CCND3, protein levels were reduced in KO1 and mRNA levels were not, consistent with a drop in CCND3 translation in these cells, but this effect was not reproduced in eIF2A KO2 (*Figure 2C–D*). Together, this indicates there is an enrichment for false-positives amongst the 15 genes identified by ribosome profiling. These results would be consistent with eIF2A having no effect on translation of any mRNA, with a few false positives coming through in the transcriptome-wide ribosome footprinting, as is always the case. Nonetheless, to test further whether translation of any of these candidates is altered upon loss of eIF2A, we generated reporter constructs by cloning the 5'-UTRs of these candidates upstream of Renilla luciferase, and then testing their expression in the presence or absence of eIF2A. We used this approach multiple times in the past to identify mRNAs whose translation is dependent on initiation factors (*Bohlen et al., 2023*; *Roiuk et al., 2024*; *Schleich et al., 2014*). We succeeded in cloning 14 of the 15 5' UTRs of the transcripts predicted to be eIF2A-dependent. None of the reporters, however, showed a consistent drop in translation in eIF2A KO1, eIF2A KO2, and siRNA-mediated eIF2A knockdown cells (*Figure 2E*, *Figure 2—figure supplement 2A*). Although we did not exclude a possible effect of eIF2A on translation via the 3'UTR or coding sequence of the 12 genes that we did not assay via immunoblotting (*Figure 2B–D*), our results indicate that eIF2A has little or no effect on translation of cellular mRNAs in HeLa cells under non-stressed conditions. Consistent with this, analysis of our ribosome footprinting data with anota2seq (*Oertlin et al., 2019*), also did not identify any transcripts with altered translation (*Figure 2—figure supplement 2C*).

Since eIF2A was proposed to deliver tRNA to the ribosome in a codon-dependent manner, we tested if loss of eIF2A affects translation initiation on a reporter where the ribosome is directly positioned on the initiation AUG. To perform this, we cloned a 5' UTR reporter with a short 5'UTR (12 nt) where loading of the small ribosomal subunit should place the AUG in close proximity to the P-site (*Gu et al., 2021*). In addition, we tested a reporter bearing the EMCV IRES in the 5'-UTR, where initiation relies on close positioning of the 40 S to the AUG (*Davies and Kaufman, 1992*). Translation of neither reporter, however, was affected by loss of eIF2A (*Figure 2—figure supplement 2D*). In sum, we were not able to find any transcript whose translation depends on eIF2A under non-stressed conditions in HeLa cells.

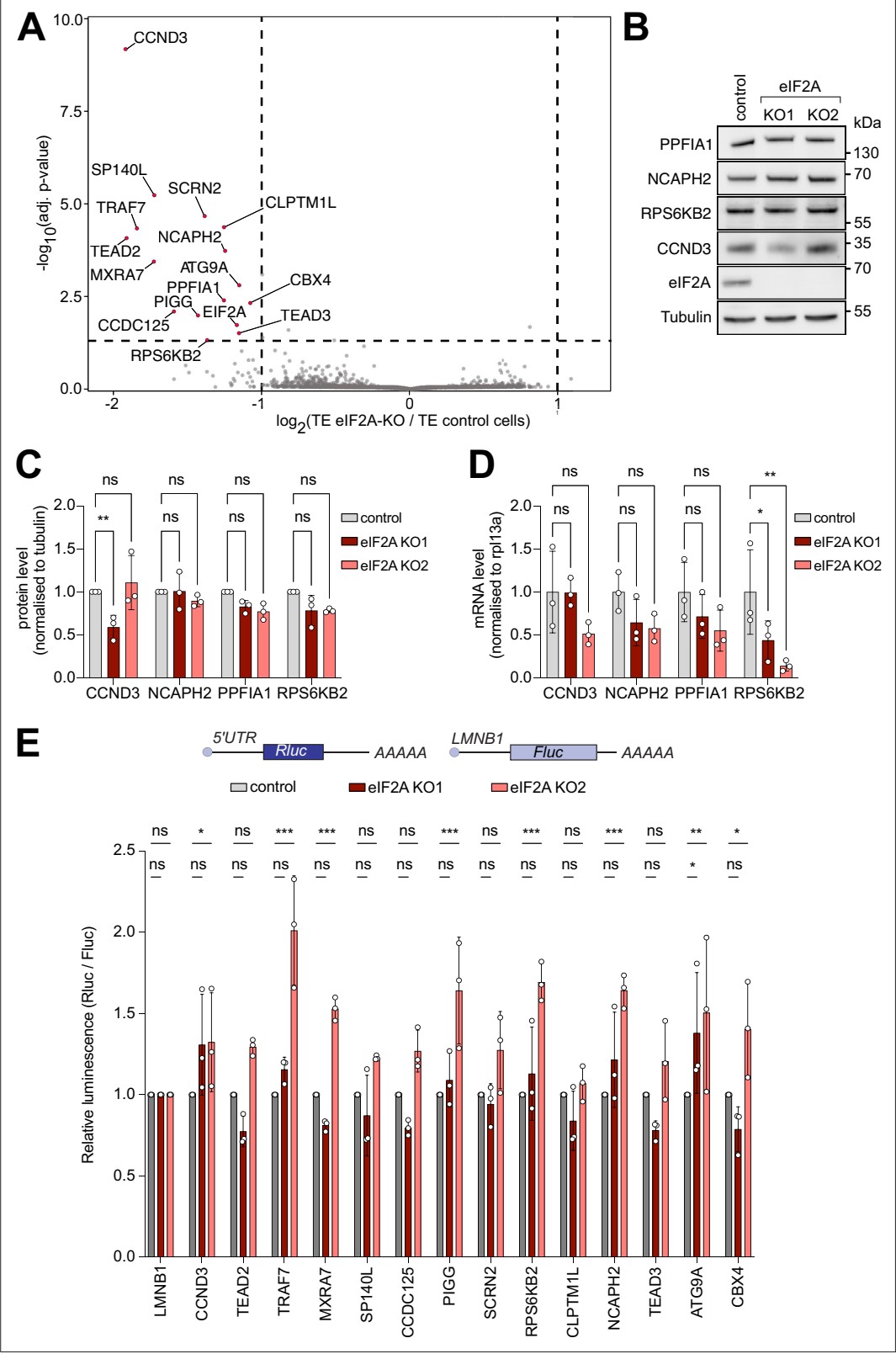

**Figure 2.** Ribosome profiling of eIF2A-KO lines finds little impact of eIF2A on translation. (**A**) Ribosome profiling identifies a handful of mRNAs sensitive to eIF2A depletion. Scatter plot of log2(fold change of Translation Efficiency eIF2A[KO]/control) versus significance. Significant candidates with log2(fold change) < −1 are shown in red. Significance was estimated with the Wald test performed by the DESeq2 package. p-values are adjusted for

*Figure 2 continued on next page*

*Figure 2 continued*

multiple comparisons.(**B–D**) Western blot validation of ribosome profiling results. Among the tested candidates, only CCND3 shows decreased protein levels in one eIF2A$^{KO}$ clone. Representative blot in (**B**), of triplicates quantified in (**C**). mRNA levels of the corresponding transcripts are quantified and shown in (**D**). Significance by Dunnett's multiple comparison test ANOVA. error bar = st. dev., ns = not significant, *p<0.05, **p<0.01. (**E**) Luciferase reporters harboring 5′ UTRs of eIF2A-dependent transcripts do not show strong changes in expression upon loss of eIF2A. Reporters carrying the 5′ UTRs of the indicated candidate genes were cloned upstream of Renilla Luciferase (RLuc) and co-transfected with a Firefly Luciferase (FLuc) normalization control. The negative control RLuc reporter and the FLuc normalization control carry the 5'UTR of Lamin B1 (LMNB1). Significance by Dunnett's multiple comparison test ANOVA. error bar = st. dev., ns = not significant, *p<0.05, **p<0.01, ***p<0.001.

The online version of this article includes the following source data and figure supplement(s) for figure 2:

**Source data 1.** pdf: Uncropped western blots.

**Source data 2.** Original immunoblot files from Chemidoc.

**Figure supplement 1.** Ribosome profiling of control and eIF2A$^{KO}$ HeLa cells.

**Figure supplement 2.** Loss of eIF2A does not affect translation of multiple different types of reporters.

**Figure supplement 2—source data 1.** pdf: Uncropped western blots.

**Figure supplement 2—source data 2.** Original immunoblot files from Chemidoc.

## eIF2A does not contribute to uORF translation

Several studies have reported that eIF2A can deliver alternative initiator tRNAs to uORFs with near-cognate start codons (*Sendoel et al., 2017*; *Starck et al., 2012*; *Starck et al., 2016*). Based on these reports, we tested if eIF2A depletion affects uORF translation in HeLa cells. First, we analyzed if translation initiation or termination on endogenous, AUG-initiated uORFs is altered in *eIF2A*$^{KO}$ cells. For this, we calculated a metagene profile of footprint reads at start and stop codons of all AUG-initiated uORFs, however, we could not detect any significant differences in uORF translation (*Figure 3—figure supplement 1A and B*).

Since uORFs are elements that inhibit translation of the downstream main ORF (mORF), initiation factors that influence uORF translation also influence, as a consequence, translation of the downstream main ORF. Therefore, we checked if translation efficiency of the main ORF of uORF-bearing transcripts changes between eIF2A$^{KO}$ and control cells. For this w,e looked at the translation efficiency of different sets of transcripts: (1) all transcripts, (2) transcripts possessing AUG-initiated uORFs, or (3) transcripts containing uORFs starting with a near cognate initiation codon – CUG, GUG, or UUG (*Figure 3—figure supplement 1C*). Although essentially no genes show a significant change in translation efficiency when analyzed singly (*Figure 2A*), this aggregate analysis might identify significant trends caused by small changes in groups of transcripts. This analysis revealed, however, that translation of transcripts with near-cognate uORFs did not differ from the global distribution in translation efficiency of all transcripts (*Figure 3—figure supplement 1C*). Transcripts with AUG-initiated uORFs, however, did have a very slight, but statistically significant increase in translation efficiency, compared to the whole dataset (log2(fold change) of –0.03 vs –0.05) (*Figure 3—figure supplement 1C*).

To validate this minor role of eIF2A in uORF translation, we tested various luciferase reporters in eIF2A$^{KO}$ cells. We placed Renilla luciferase under control of a 5′ UTR with no uORF (LMNB1, as a negative control) or the same 5'UTR where we synthetically introduced a uORF with different start codons or Kozak sequences (*Figure 3A*). In line with our ribosome profiling data, all of these luciferase reporters showed no significant difference in expression in *eIF2A*$^{KO}$ or eIF2A overexpressing cells compared to controls (*Figure 3A*, *Figure 3—figure supplement 1D*). Considering that this luciferase assay is a readout for translation of the main ORF, and not the uORF directly, we also performed an experiment where we directly visualized uORF peptide production. For this purpose, we generated fluorescent reporters carrying a uORF coding for the SINFEKL peptide, which can be presented on the surface of cells expressing the H-2Kb Class I MHC complex (HEK293T-H2-Kb) where it can be detected with antibodies (*Starck et al., 2016*). To assess the impact of eIF2A on the production of this peptide, we generated a HEK293T-H2-Kb, *eIF2A*$^{KO}$ cell line (*Figure 3B*). Like *eIF2A*$^{KO}$ HeLa cells, HEK293T-H2-Kb *eIF2A*$^{KO}$ cells also did not display any defect in proliferation (*Figure 3—figure supplement 1E*) or global translation (*Figure 3—figure supplement 1F–G*). This revealed, however, no difference in

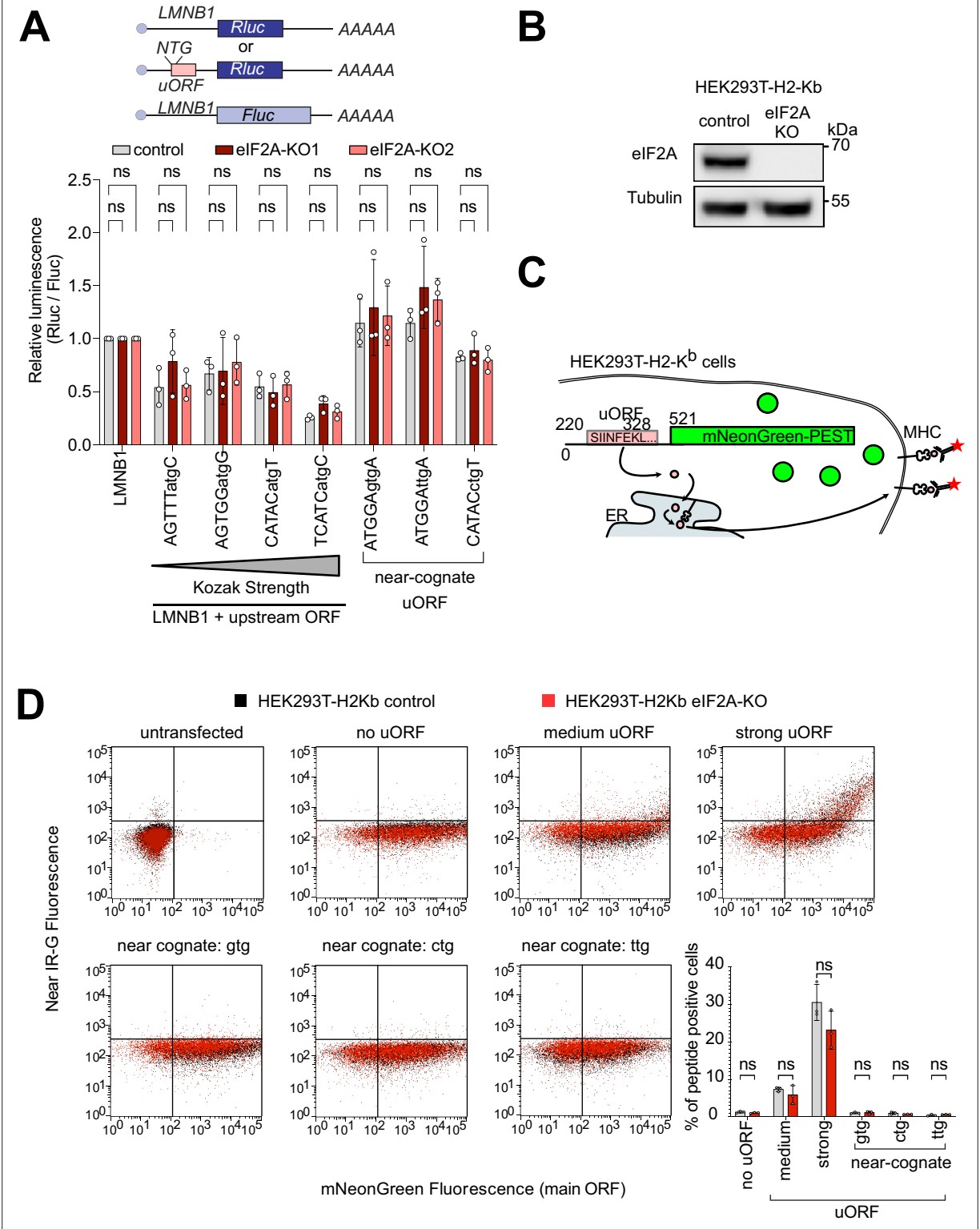

**Figure 3.** eIF2A has little or effect on uORF translation. (**A**) Synthetic reporters harboring uORFs with different start codons and initiation contexts do not show dependence on eIF2A. The sequence context of the uORF start codons is indicated: either AUG or a near-cognate start codon (GTG, TTG, CTG) was used. Significance by Dunnett's multiple comparison test ANOVA, error bar = st. dev. ns = not significant. (**B**) Validation that HEK293T-H2-K^b *eIF2A^KO* cells have no eIF2A protein by immunoblotting. (**C**) Schematic diagram illustrating the setup to simultaneously detect a small peptide produced by a uORF and fluorescent mNeonGreen encoded by the main ORF. The short peptide SIINFEKL is presented on the cell surface by MHC-I and

*Figure 3 continued on next page*

*Figure 3 continued*

detected using a monoclonal antibody. (**D**) *eIF2A* knockout does not cause a drop in uORF translation. In the graph to the right, the percent of uORF-positive cells relative to all mNeonGreen cells is quantified. Significance by unpaired, two-sided, t-test. ns = not significant.

The online version of this article includes the following source data and figure supplement(s) for figure 3:

**Source data 1.** pdf: Uncropped western blots.

**Source data 2.** Original immunoblot files from Chemidoc.

**Figure supplement 1.** Knockout of *eIF2A* has no effect on uORF translation.

**Figure supplement 1—source data 1.** pdf: Uncropped western blots.

**Figure supplement 1—source data 2.** Original immunoblot files from Chemidoc.

translation of the uORF peptide when comparing *eIF2A*^KO to control cells, regardless of whether the uORF was initiated by an AUG codon or a near-cognate initiation codon, as detected by flow cytometry (*Figure 3D*). This differs from what we previously observed upon loss of PRRC2A/B/C proteins, which caused increased expression of the uORF SINFEKL peptide and reduced expression of the downstream main ORF (*Bohlen et al., 2023*).

## No detectable role for eIF2A in translation when eIF2 is inhibited

eIF2A has been proposed to act as a backup pathway for tRNA delivery when eIF2 function is attenuated due to phosphorylation of its alpha subunit by one of the kinases of the integrated stress response (*Kim et al., 2018*; *Kwon et al., 2017*; *Starck et al., 2016*; *Tusi et al., 2021*). We, therefore, induced the integrated stress response and tested the impact of loss of eIF2A on several different translational changes that occur – the global reduction in translation levels, the formation of stress granules, and induction of the few target genes such as ATF4 which evade the global reduction in translation and instead are translationally induced (*Andreev et al., 2015*; *Sidrauski et al., 2015*). For this, we treated cells with tunicamycin or sodium arsenite to phosphorylate eIF2α (eIF2S1) by two independent kinases – PERK or HRI – and estimated global translation by assessing polysome profiles. Both treatments led to suppression of translation (compared *Figure 4—figure supplement 1A–B* to *Figure 1E*), however, the magnitude of suppression was similar in *eIF2A*^KO and isogenic control HeLa cells (*Figure 4—figure supplement 1A–B*). This suggests that, as in non-stressed conditions, eIF2A has a minimal effect on global translation also when the integrated stress response is active. We next tested if *eIF2A*^KO cells have any defect in stress granule formation, which is a hallmark of translation inhibition upon eIF2α phosphorylation (*Sidrauski et al., 2015*). However, *eIF2A*^KO cells formed stress granules to the same degree as the isogenic control HeLa cells (*Figure 4—figure supplement 1C–D*). To test the induction of target genes during the integrated stress response we selected tunicamycin treatment because sodium arsenite is too strong and completely shuts off all translation. We cloned luciferase reporters carrying 5'UTRs of genes that were previously shown to be translationally upregulated upon eIF2α phosphorylation (*Andreev et al., 2015*). We then transfected these into control or *eIF2A*^KO HeLa cells and treated the cells with either DMSO or 1 µg/ml tunicamycin for 16 hr. Tunicamycin treatment resulted in the same level of eIF2α phosphorylation in either cell line (*Figure 4—figure supplement 1E*). None of the reporters we tested – ATF4, PPP1R15B, IFRD1 – had an induction defect in *eIF2A*^KO cells (*Figure 4A*). This is in line with a previous report showing that in yeast, the ATF4 homolog GCN4 is induced properly during amino acid starvation in cells lacking eIF2A (*Zoll et al., 2002*).

The assays described above mainly assess bulk translation, and might have missed changes in translation of individual mRNAs. To test whether eIF2A affects translation of any mRNA when eIF2 is inactive, we performed ribosome profiling on control and *eIF2A*^KO HeLa cells treated with tunicamycin and compared these data to the data from non-stressed controls described above. The results showed good reproducibility across duplicates for both ribosome profiling and total RNA samples (*Figure 4—figure supplement 2A*). Surprisingly, by comparing translation efficiency for all mRNAs in *eIF2A*^KO versus control cells in the tunicamycin-treated condition, we did not find any transcript with significantly changed translation apart from eIF2A itself (*Figure 4—figure supplement 2B*). In line with this, we also did not find any significant changes in a metagene profile of footprints on all main ORFs in control and *eIF2A*^KO cells upon tunicamycin treatment (*Figure 4—figure supplement 2C–D*).

Nonetheless, it is possible that the induction of particular transcripts in response to stress may be affected in eIF2A-knock-out cells compared to controls. To test it, we compared changes in translation

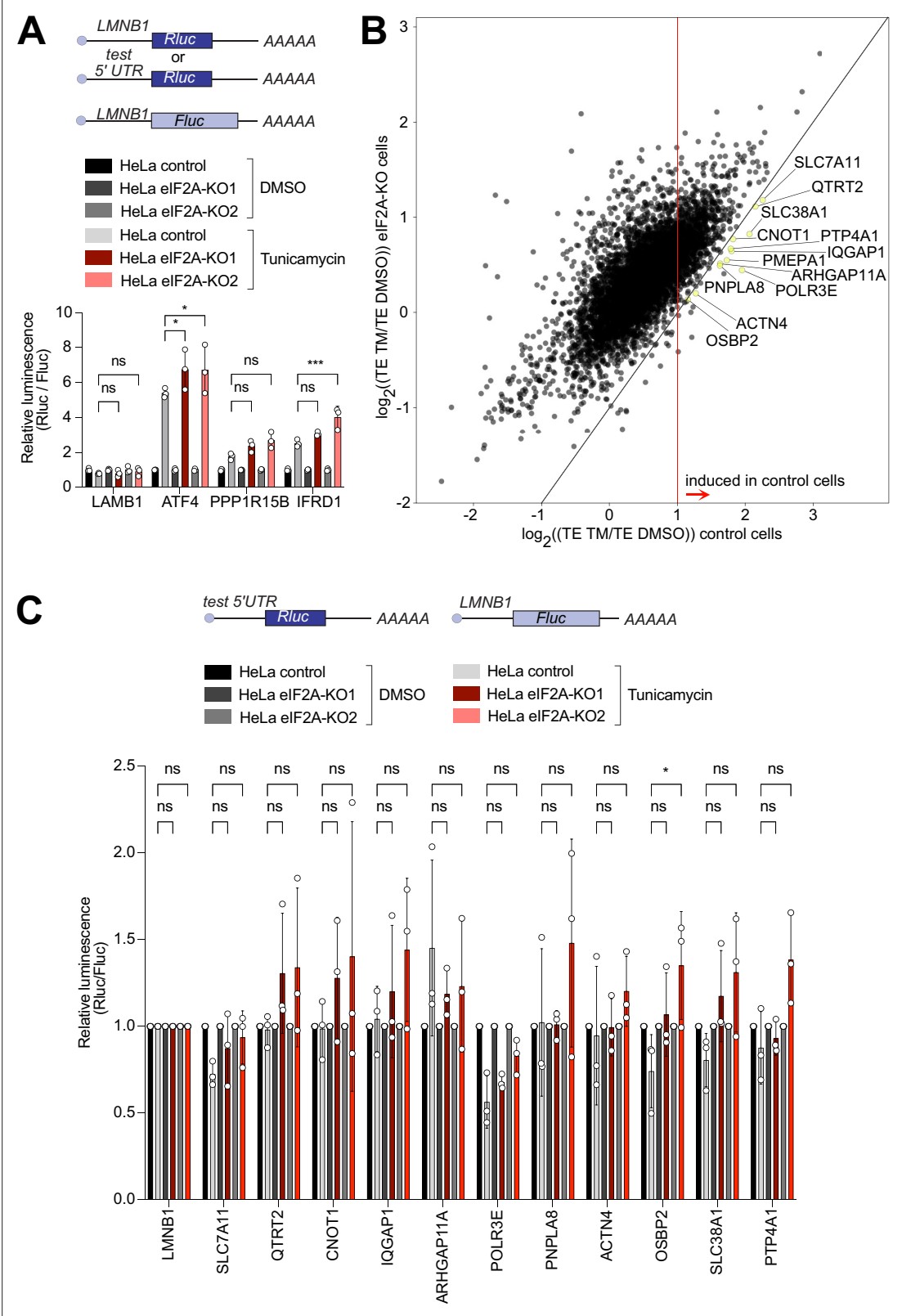

**Figure 4.** eIF2A has a minor impact on translation during the integrated stress response. (**A**) Loss of eIF2A does not blunt induction of target genes of the integrated stress response. Reporters carrying the 5' UTRs of the indicated candidate genes were co-transfected with a Firefly Luciferase (FLuc) normalization control reporter into eIF2A^KO and control HeLa cells and treated for 16 hr either with DMSO or 1 μg/ml tunicamycin (TM). Significance by Dunnett's multiple comparison test ANOVA. error bar = st. dev., ns = not significant, *p<0.05, ***p<0.001 (**B**) Ribosome profiling identifies 12 mRNAs

*Figure 4 continued on next page*

*Figure 4 continued*

that are significantly induced upon tunicamycin treatment (1 ug/ml) in control cells but not eIF2A$^{KO}$ cells. Scatter plot of log2(fold change) of Translation Efficiency TM/DMSO for control cells on the x-axis versus eIF2A$^{KO}$ cells on the y-axis. mRNAs that are statistically significantly induced with log2(fold change)>1 in control cells but not in eIF2A$^{KO}$ cells are shown in yellow and marked by gene name. Significance was estimated with the Wald test performed by DESeq2 thepackage. p-values are adjusted for multiple comparisons. (**C**) Transfection of luciferase reporters harboring 5' UTRs of eIF2A-dependent transcripts does not show impaired induction in eIF2A$^{KO}$ cells upon tunicamycin treatment. 5' UTRs of eIF2A-dependent transcripts from panel B were cloned upstream of Renilla luciferase and co-transfected with a FLuc normalization control reporter into control or EIF2A$^{KO}$ HeLa cells with subsequent treatment for 16 hr either with DMSO or 1 μg/ml TM. Significance by Dunnett's multiple comparison test ANOVA. error bar = st. dev., ns = not significant, *p<0.05.

The online version of this article includes the following source data and figure supplement(s) for figure 4:

**Figure supplement 1.** The integrated stress response suppresses translation equally well in control and *eIF2A*$^{KO}$ HeLa cells.

**Figure supplement 1—source data 1.** pdf: Uncropped western blots.

**Figure supplement 1—source data 2.** Original immunoblot files from Chemidoc.

**Figure supplement 2.** Ribosome profiling of tunicamycin-treated *eIF2A*$^{KO}$ and control HeLa cells.

**Figure supplement 3.** Translation of uORF-bearing transcripts is not affected upon loss of eIF2A in tunicamycin-treated cells.

**Figure supplement 4.** Expression of initiation factors reported to possess tRNA$_i$$^{Met}$ binding activities.

efficiency between untreated and treated cells in both control and *eIF2A*$^{KO}$ cells. This identified a small group of transcripts whose translation was more strongly induced in control cells than in *eIF2A*$^{KO}$ cells (***Figure 4B***). Statistical analysis identified 12 transcripts that were significantly induced in control cells (log2 fold change of TE (TM/untreated) ≥ 1, p-adj <0.05), but with blunted induction in *eIF2A*$^{KO}$ cells (yellow dots, ***Figure 4B***). Translational induction of genes in response to eIF2α phosphorylation is thought to occur mainly via their 5'UTRs. Therefore, to study systematically the induction of these 12 transcripts, we cloned their 5'-UTRs into luciferase reporters. We succeeded in cloning the 5'UTRs of 11 out of the 12 transcripts. Transfection of these reporters and co-treatment with tunicamycin, however, did not show any impaired induction in *eIF2A*$^{KO}$ cells compared to control cells (***Figure 4C***). This suggests that either the induction in translation of these transcripts relies on features outside of their 5'-UTRs, or these 12 transcripts are false positives from the ribosome footprinting.

Finally, we assessed if there is any impact of eIF2A loss on translation of uORF-containing transcripts upon tunicamycin treatment. First, we looked at the metagene profiles for footprints on all uORFs aligned to their start or stop codons, however, this did not show any significant changes, apart from a minor faster transition from initiation to elongation in eIF2A$^{KO}$ cells (***Figure 4—figure supplement 3A–B***), which is the opposite of what one would expect. Next, we analyzed how the translation efficiency of transcripts containing either AUG- or near-cognate-initiated uORFs changes upon loss of eIF2A, compared to all transcripts, however, we did not find any significant changes in either case (***Figure 4—figure supplement 3C***). Lastly, we tested the synthetic reporters containing uORFs with different start codons or Kozak sequences, described above, in the presence of tunicamycin but found no differences in translation in eIF2A$^{KO}$ cells compared to controls (***Figure 4—figure supplement 3D***). In sum, overall, we find a very minor, or no contribution of eIF2A on translation upon stress in HeLa cells.

## Discussion

Although eIF2A was discovered prior to eIF2α, its role in translation still remains unclear, with a broad range of different functions attributed to it (reviewed in ***Komar and Merrick, 2020***). In this study, we aimed to systematically assess the impact of eIF2A on translation regulation in HeLa cells. Our data show that more eIF2A is localized to the cytosol than the nucleus. Despite a previous report that eIF2A shuttles out of the nucleus in response to cellular stresses (***Kim et al., 2011***), we observed no change in its distribution between these compartments upon stress, consistent with previous findings in HAP1 cells (***González-Almela et al., 2018***; ***Sanz et al., 2017***). Recently, eIF2A was reported to act as a global suppressor of translation, affecting all mRNAs (***Grove et al., 2023***). This would result in increased global translation in *eIF2A* knock-out cells. However, we did not observe any change in bulk translation, as measured by OPP incorporation assays, upon loss or increased expression of eIF2A. In line with no impact of eIF2A on global translation, we also did not detect any proliferation

defect of eIF2A^KO cells, which fits with a dispensable role of eIF2A in other species (*Anderson et al., 2021*; *Golovko et al., 2016*; *Kim et al., 2018*; *Komar et al., 2005*). This is further supported by our ribosome profiling data, which showed that only a few transcripts, if any, changed in their translation upon loss of eIF2A, while the vast majority of the translatome remained unaffected. Given that eIF2A was reported to modulate translation initiation through elements in the 5'-UTR (e.g. uORFs, repeats, etc.) we tested whether cloning the 5'-UTRs of affected transcripts into luciferase reporters would reveal eIF2A dependence, however, this was not the case. This suggests that if certain transcripts are affected by the lack of eIF2A this is not due to their 5' UTRs and may rely on features in their CDS or 3'-UTRs.

The tRNA-binding properties of eIF2A remain a topic of debate. While initial studies reported that eIF2A could bind tRNA^Met in a GTP-independent manner (*Kim et al., 2018*; *Zoll et al., 2002*), the purification protocols used in these studies were later shown to result in eIF2D contamination, which was subsequently identified as the tRNA^Met binding factor, whereas eIF2A showed no such affinity (*Dmitriev et al., 2010*). Despite the contested tRNA-binding capabilities of eIF2A, several subsequent studies still reported a switch of tRNA delivery from eIF2α to eIF2A upon cellular stresses, when eIF2 activity is attenuated due to phosphorylation by one of the four ISR kinases (*Kim et al., 2018*; *Kwon et al., 2017*; *Starck et al., 2016*; *Tusi et al., 2021*). Taking this into consideration, we carried out ribosome profiling upon tunicamycin treatment, which results in eIF2α phosphorylation through PERK kinase. Surprisingly, also in this condition, we did not find any transcript whose translation is significantly affected when comparing eIF2A^KO to control cells. However, we did find some transcripts whose induction was blunted in eIF2A^KO cells. Since no transcript showed reduced translation in eIF2A^KO cells compared to control cells in either the control or the tunicamycin condition, these must be transcripts which had mildly, but not significantly increased translation in the non-stressed condition and mildly, but not significantly reduced translation in the tunicamycin condition, leading to a significant difference when comparing the two treatment conditions. To further dissect if this defect in induction arises from elements within the transcript 5'-UTRs, we cloned their 5'UTRs into luciferase reporters, however, we found no significant difference between eIF2A^KO and control cells. Given that the more straightforward analysis comparing translation efficiency in eIF2A^KO versus control cells in the tunicamycin condition revealed no transcripts with altered translation, we think the most simple explanation is that also in the stressed condition where eIF2 function is suppressed, eIF2A does not impact translation in HeLa cells.

Several reports linked eIF2A function to uORF translation. Although our ribosome profiling data indicates that there are no obvious defects in translation of transcripts with uORFs, we nonetheless decided to test the impact of eIF2A on synthetic uORF reporters. We used reporters with uORFs initiated by either AUG or by near-cognate start codons, thereby testing the role of eIF2A in leaky scanning, reinitiation, and near-cognate initiation. Our data, however, show that none of the tested reporters was affected by eIF2A loss or overexpression in both non-stressed and stressed conditions, indicating that eIF2A is dispensable for uORF translation in HeLa cell lines. We did not detect increased translation (i.e. footprints) of any other initiation factor that can potentially deliver tRNAs in *eIF2A* knock-outs (*Figure 4—figure supplement 4*), however we cannot exclude that the functional consequence of eIF2A loss is masked by another protein with redundant function.

Overall, our findings are fully in agreement with recent reports showing little or no effect on translation of eIF2A loss in yeast or human HEK293-T cells (*Gaikwad et al., 2024*; *Ichihara et al., 2021*). Ichihara and colleagues generated eIF2A knockout HEK293-T cells and compared their translation to parental control cells via ribosome profiling. This revealed only 1 mRNA with reduced translation efficiency in non-stressed conditions and 4 mRNAs with reduced translation efficiency in cells treated with arsenite where eIF2 is inhibited (*Ichihara et al., 2021*). Thus, the lack of impact of eIF2A on mRNA translation does not appear to be specific to HeLa cells.

Historically, eIF2A was linked to translation because it was purified with other initiation factors and it shows synthetic lethality with eIF4E (*Komar et al., 2005*). Even though some of the initial tests with eIF2A showed no activity in reconstituted translational extracts (*Merrick and Anderson, 1975*), subsequent reports attributed different translational functions to eIF2A. Given that eIF2A contains an RNA-binding domain, it is possible that eIF2A plays a role in RNA trafficking or mRNA decay. The synthetic lethality between eIF2A and eIF4E (*Komar et al., 2005*) is interesting in this regard since eIF4E is also involved in the nuclear export of transcripts with 4E-SE elements. This raises the

possibility that eIF2A might contribute to mRNA regulation beyond translation initiation. For instance, the last 50 amino acids of eIF2A are highly similar to PYM (*Diem et al., 2007*), a protein that binds to the 40 S ribosomal subunit and to Y14, an exon-junction complex protein.

In this study, we used predominantly HeLa cells, with some tests in HEK293T cells. Thus, we cannot rule out that in other cell types, eIF2A might have a function in translation initiation. eIF2A knockout mice have a reduced abundance of B-lymphocytes and dendritic cells in the thymic medulla, as well as lipid metabolism defects (*Anderson et al., 2021*). A recent study reported that mutation of *eIF2A* in *Drosophila melanogaster* via a MiMic transposon insertion into the second exon of *eIF2A* is lethal for the organism (*Lowe and Montell, 2022*). This is the first example of a lethal phenotype for eIF2A-loss in an organism, although this result needs to be confirmed with a clean knockout. The same study found that insertion of a different, piggyback transposon into the second intron resulted in viable, but infertile males due to failed sperm individualization, supporting the idea of cell-type specific function of eIF2A (*Lowe and Montell, 2022*). Whether these phenotypes in *Drosophila* are due to a role of eIF2A in translation initiation, however, remains to be investigated. Overall, our results support the idea that eIF2A plays a minor, or no role in regulating translation initiation in human HeLa and HEK293T cells.

## Materials and methods

### Key resources table

| Reagent type (species) or resource | Designation | Source or reference | Identifiers | Additional information |
|---|---|---|---|---|
| Gene (*Homo sapiens*) | eIF2A | Ensemble | ENSG00000144895 | |
| Cell line (*Homo sapiens*) | HeLa | DSMZ | ACC 57 | |
| Cell line (*Homo sapiens*) | HeLa-eIF2A-KO | This study | This study | |
| Cell line (*Homo sapiens*) | HEK293T-H2-Kb | gift from Rienk Offringa lab | Rienk Offringa lab | |
| Cell line (*Homo sapiens*) | HEK293T-H2-Kb-eIF2A-KO | This study | This study | |
| Transfected construct (human) | | | D-014766–01 | GCUCCCAGGUUACGGGUUA |
| Transfected construct (human) | | | D-014766–02 | GAUUUGGAAUUGGGUAUUU |
| Transfected construct (human) | | | D-014766–03 | GCAGAUAAAGUUACAAUGC |
| Transfected construct (human) | siRNA to eIF2A (mix of all 4 was used) | Horizon discovery | D-014766–04 | CCACAAUCAGGAAACGAUA |
| Sequence-based reagent (oligos used to produce KO) | Pair 8 sg_eIF2A 1 | This study | OMR064 | CACCGCTCACCCAA AAATACTGTCC |
| Sequence-based reagent (oligos used to produce KO) | Pair 8 sg_eIF2A 2 | This study | OMR065 | AAACGGACAGTATTTT TGGGTGAGC |
| Sequence-based reagent (oligos used to produce KO) | Pair 5 sg_eIF2A 1 | This study | OMR058 | CACCGAATACTAATATA TGTCCATG |
| Sequence-based reagent (oligos used to produce KO) | Pair 5 sg_eIF2A 2 | This study | OMR059 | AAACCATGGACATATAT TAGTATTC |
| Antibody | anti-ATF-4 (D4B8) (Rabbit monoclonal) | Cell Signaling | cat. No #11815 Lot#6 | WB (1:1000) |
| Antibody | anti-C-Myc (Rabbit monoclonal) | Cell Signaling | cat. No. #13987 Lot#6 | WB (1:1000) |
| Antibody | anti-CCND3 (Rabbit polyclonal) | invitrogen | cat no. #PA5-80416 Lot#UH2828593 | WB (1:1000) |
| Antibody | anti-eIF2A (3A7A8) (mouse monoclonal) | santa cruz | cat. No. sc-517214 Lot#B0821 | WB (1:1000) |
| Antibody | anti-FLAG (Rabbit polyclonal) | SIGMA | cat. No. F7425-.2MG Lot#0000252651 | WB (1:1000) |
| Antibody | anti-GAPDH (Rabbit monoclonal) | Cell Signaling | cat. No. #2118 LOT#16 | WB (1:1000) |
| Antibody | anti-HSP90 (Rabbit monoclonal) | Cell signaling | cat. No. 4877 Lot#6 | WB (1:1000) |

*Continued on next page*

*Continued*

| Reagent type (species) or resource | Designation | Source or reference | Identifiers | Additional information |
|---|---|---|---|---|
| Antibody | anti-Lamin A/C (636) (mouse monoclonal) | Santa Cruz | cat. No. sc-7292 Lot#C0218 | WB (1:1000) |
| Antibody | anti-NCAPH2 (Rabbit polyclonal) | Proteintech | cat. No. 26172–1-AP Lot#00039440 | WB (1:1000) |
| Antibody | anti-p-p38 (Rabbit polyclonal) | Cell Signaling | cat. No. #9211 Lot#25 | WB (1:1000) |
| Antibody | anti-PPFIA1 (Rabbit polyclonal) | Proteintech | cat. No. 14175–1-AP Lot#00005224 | WB (1:1000) |
| Antibody | anti-puromycin (mouse monoclonal) | Sigma | cat. No. MABE343 Lot#3484967 | WB (1:1000) |
| Antibody | anti-RPS6KB2 (Rabbit polyclonal) | Proteintech | cat. No. 26194–1-AP Lot#00040692 | WB (1:1000) |
| Antibody | anti-Tubulin | Sigma | cat. No. T9026 LOT#0000307925 | WB (1:2500) |
| Antibody | anti-G3BP1 (mouse monoclonal) | santa cruz | cat. No. sc-81940 Lot@G0617 | IF: 1:50 |
| Sequence-based reagent (plasmids with 5' UTR of interest cloned upstream of renilla luciferase) | ACTN4 | This study | pMR1184 | 5' UTR cloned from transcript with id ENST00000252699 |
| Sequence-based reagent (plasmids with 5' UTR of interest cloned upstream of renilla luciferase) | ARHGAP11A | This study | pMR181 | 5' UTR cloned from transcript with id ENST00000361627 |
| Sequence-based reagent (plasmids with 5' UTR of interest cloned upstream of renilla luciferase) | ATG9A | This study | pMR091 | 5' UTR cloned from transcript with id ENST00000361242 |
| Sequence-based reagent (plasmids with 5' UTR of interest cloned upstream of renilla luciferase) | CBX4 | This study | pMR1171 | 5' UTR cloned from transcript with id ENST00000269397 |
| Sequence-based reagent (plasmids with 5' UTR of interest cloned upstream of renilla luciferase) | CCDC125 | This study | pMR1174 | 5' UTR cloned from transcript with id ENST00000383374 |
| Sequence-based reagent (plasmids with 5' UTR of interest cloned upstream of renilla luciferase) | CCND3 | This study | pMR965 | 5' UTR cloned from transcript with id ENST00000372991 |
| Sequence-based reagent (plasmids with 5' UTR of interest cloned upstream of renilla luciferase) | CLPTM1L | This study | pMR1178 | 5' UTR cloned from transcript with id ENST00000337392 |
| Sequence-based reagent (plasmids with 5' UTR of interest cloned upstream of renilla luciferase) | CNOT1 | This study | pMR1182 | 5' UTR cloned from transcript with id ENST00000317147 |
| Sequence-based reagent (plasmids with 5' UTR of interest cloned upstream of renilla luciferase) | IQGAP1 | This study | pMR168 | 5' UTR cloned from transcript with id ENST00000268182 |
| Sequence-based reagent (plasmids with 5' UTR of interest cloned upstream of renilla luciferase) | MXRA7 | This study | pMR1173 | 5' UTR cloned from transcript with id ENST00000355797 |
| Sequence-based reagent (plasmids with 5' UTR of interest cloned upstream of renilla luciferase) | NCAPH2 | This study | pMR1170 | 5' UTR cloned from transcript with id ENST00000420993 |
| Sequence-based reagent (plasmids with 5' UTR of interest cloned upstream of renilla luciferase) | OSBP2 | This study | pMR664 | 5' UTR cloned from transcript with id ENST00000332585 |
| Sequence-based reagent (plasmids with 5' UTR of interest cloned upstream of renilla luciferase) | PIGG | This study | pMR096 | 5' UTR cloned from transcript with id ENST00000310340 |
| Sequence-based reagent (plasmids with 5' UTR of interest cloned upstream of renilla luciferase) | PNPLA8 | This study | pMR1183 | 5' UTR cloned from transcript with id ENST00000257694 |
| Sequence-based reagent (plasmids with 5' UTR of interest cloned upstream of renilla luciferase) | POLR3E | This study | pMR1181 | 5' UTR cloned from transcript with id ENST00000640588 |

*Continued on next page*

*Continued*

| Reagent type (species) or resource | Designation | Source or reference | Identifiers | Additional information |
|---|---|---|---|---|
| Sequence-based reagent (plasmids with 5' UTR of interest cloned upstream of renilla luciferase) | PTP4A1 | This study | pMR196 | 5' UTR cloned from transcript with id ENST00000626021 |
| Sequence-based reagent (plasmids with 5' UTR of interest cloned upstream of renilla luciferase) | QTRT2 | This study | pMR1180 | 5' UTR cloned from transcript with id ENST00000281273 |
| Sequence-based reagent (plasmids with 5' UTR of interest cloned upstream of renilla luciferase) | RPS6KB2 | This study | pMR1176 | 5' UTR cloned from transcript with id ENST00000312629 |
| Sequence-based reagent (plasmids with 5' UTR of interest cloned upstream of renilla luciferase) | SCRN2 | This study | pMR1175 | 5' UTR cloned from transcript with id ENST00000290216 |
| Sequence-based reagent (plasmids with 5' UTR of interest cloned upstream of renilla luciferase) | SLC38A1 | This study | pMR1185 | 5' UTR cloned from transcript with id ENST00000546893 |
| Sequence-based reagent (plasmids with 5' UTR of interest cloned upstream of renilla luciferase) | SLC7A11 | This study | pMR1179 | 5' UTR cloned from transcript with id ENST00000280612 |
| Sequence-based reagent (plasmids with 5' UTR of interest cloned upstream of renilla luciferase) | SP140L | This study | pMR1177 | 5' UTR cloned from transcript with id ENST00000415673 |
| Sequence-based reagent (plasmids with 5' UTR of interest cloned upstream of renilla luciferase) | TEAD2 | This study | pMR1172 | 5' UTR cloned from transcript with id ENST00000311227 |
| Sequence-based reagent (plasmids with 5' UTR of interest cloned upstream of renilla luciferase) | TEAD3 | This study | pMR1169 | 5' UTR cloned from transcript with id ENST00000338863 |
| Sequence-based reagent (plasmids with 5' UTR of interest cloned upstream of renilla luciferase) | TRAF7 | This study | pMR967 | 5' UTR cloned from transcript with id ENST00000326181 |
| Sequence-based reagent (qRT-PCR oligos) | eIF2A qRT-PCR | This study | OMR202 | AAAGCACAGTGTTTCCAAGGG |
| Sequence-based reagent (qRT-PCR oligos) | eIF2A qRT-PCR | This study | OMR203 | GCAGTAGTCCCTTGTTAGTGA |
| Sequence-based reagent (qRT-PCR oligos) | CCND3 qRT-PCR oligo | This study | OMR412 | GAAGGGGCGTCTGTTCC |
| Sequence-based reagent (qRT-PCR oligos) | CCND3 qRT-PCR oligo | This study | OMR413 | CAGGGAGGAGGAGCTTG |
| Sequence-based reagent (qRT-PCR oligos) | NCAPH2 qRT-PCR oligo | This study | OMR2542 | CGAGTATCTGGAGGAGCTGGATCA |
| Sequence-based reagent (qRT-PCR oligos) | NCAPH2 qRT-PCR oligo | This study | OMR2543 | GCCTGGTAGACGAGTGAGTAGAGG |
| Sequence-based reagent (qRT-PCR oligos) | PPFIA1 qRT-PCR oligo | This study | OMR2546 | AGCAGAAAGGAATAACACCAGGCT |
| Sequence-based reagent (qRT-PCR oligos) | PPFIA1 qRT-PCR oligo | This study | OMR2547 | CATCCAGAGCTTTGTGGTGTTCAA |
| Sequence-based reagent (qRT-PCR oligos) | RPS6KB2 qRT-PCR oligo | This study | OMR2550 | TGGATTTGGAGACGGAGGAAGGCA |
| Sequence-based reagent (qRT-PCR oligos) | RPS6KB2 qRT-PCR oligo | This study | OMR2551 | GATGCGCTCTGGGCCAACGTTCAC |
| Sequence-based reagent (qRT-PCR oligos) | RPL13A qRT-PCR oligo | This study | OMR068 | CCGCCCTACGACAAGAAA |
| Sequence-based reagent (qRT-PCR oligos) | RPL13A qRT-PCR oligo | This study | OMR069 | CAGGGTGGCTGTCACTGC |
| Sequence-based reagent (qRT-PCR oligos) | GAPDH q-RT-PCR | This study | OMR496 | CCTTTGACGCTGGGGCT |
| Sequence-based reagent (qRT-PCR oligos) | GAPDH q-RT-PCR | This study | OMR497 | GGTGGTCCAGGGGTCTT |
| Sequence-based reagent (oligos used to clone 5'UTR of interest) | OSBP2 | This study | OMR1589 | ccggaagcttACTGGCCGCTCGGCCGCGCGCGGGTCGGCCGGCTCTccaccATGacTTCGAAccgg |
| Sequence-based reagent (oligos used to clone 5'UTR of interest) | OSBP2 | This study | OMR1590 | ccggTTCGAAgtCATggtggAGAGCCGGCCGACCCGCGCGCGGCCGAGCGGCCAGTaagcttccgg |
| Sequence-based reagent (oligos used to clone 5'UTR of interest) | ctg uORF | This study | OMR1877 | ccggaCATACctgTatTCGATAATCAACTTTGAAAAACTCtaaa |

*Continued on next page*

*Continued*

| Reagent type (species) or resource | Designation | Source or reference | Identifiers | Additional information |
|---|---|---|---|---|
| Sequence-based reagent (oligos used to clone 5'UTR of interest) | ctg uORF | This study | OMR1878 | CCGGtttaGAGTTTTTCAAA GTTGATTATCGAatAcagGTATGt |
| Sequence-based reagent (oligos used to clone 5'UTR of interest) | gtg uORF | This study | OMR1879 | ccggaTGGAgtgAaaTCGATA ATCAACTTTGAAAAACTCtaaa |
| Sequence-based reagent (oligos used to clone 5'UTR of interest) | gtg uORF | This study | OMR1880 | CCGGtttaGAGTTTTTCAAAGT TGATTATCGAatAcacGTATGt |
| Sequence-based reagent (oligos used to clone 5'UTR of interest) | ttg uORF | This study | OMR1881 | ccggaTGGAttgAaaTCGATAA TCAACTTTGAAAAACTCtaaa |
| Sequence-based reagent (oligos used to clone 5'UTR of interest) | ttg uORF | This study | OMR1882 | CCGGtttaGAGTTTTTCAAAGT TGATTATCGAatAcaaGTATGt |
| Sequence-based reagent (oligos used to clone 5'UTR of interest) | TRAF7 | This study | OMR2182 | ccggaagcttGGCAGCCGTCCGGGC |
| Sequence-based reagent (oligos used to clone 5'UTR of interest) | TRAF7 | This study | OMR2183 | ccggTTCGAAgtCATggtggGCTCTA GAGAGGCATCTACGGTCCTT |
| Sequence-based reagent (oligos used to clone 5'UTR of interest) | TEAD2 | This study | OMR2504 | AGCttCCCACTTTTCCCAAACA AAGCTCCCGGCAACTTTCTCC CTCGCAGCGCCCCGCCCGCC CGCGGCTCCCCAGCCCCAGGC CGGGAGGCCCAGcCATGACTT |
| Sequence-based reagent (oligos used to clone 5'UTR of interest) | TEAD2 | This study | OMR2505 | CGAAGTCATGgCTGGGCCT CCCGGCCTGGGGCTGGGG AGCCGCGGGCGGGCGGGG CGCTGCGAGGGAGAAAGTT GCCGGGAGCTTTGTTTGGGA AAAGTGGGa |
| Sequence-based reagent (oligos used to clone 5'UTR of interest) | MXRA7 | This study | OMR2506 | agcttACTCGGCGGCC GCGGCGCGccatgactt |
| Sequence-based reagent (oligos used to clone 5'UTR of interest) | MXRA7 | This study | OMR2507 | cgaagtcatggCGCGCCG CGGCCGCCGAGTa |
| Sequence-based reagent (oligos used to clone 5'UTR of interest) | CCDC125 | This study | OMR2508 | agcttGCGGCGGCAGCGGCG CACGCGCACGGAGAGGAGG CTACTTGCCAGACAGCCCATT TTTTCTTATGATAAAGACGGCA TTTGGCTCccatgactt |
| Sequence-based reagent (oligos used to clone 5'UTR of interest) | CCDC125 | This study | OMR2509 | cgaagtcatggGAGCCAAATGC CGTCTTTATCATAAGAAAAAA TGGGCTGTCTGGCAAGTAGC CTCCTCTCCGTGCGCGTGCG CCGCTGCCGCCGCa |
| Sequence-based reagent (oligos used to clone 5'UTR of interest) | SCRN2 | This study | OMR2510 | agcttGCGGCCCTGGCCAGAAG CGGAGGAGGTGGCACCCGGG ACCGAGCTGGGGTCTTGGAGG AAGAGAGGccatgactt |
| Sequence-based reagent (oligos used to clone 5'UTR of interest) | SCRN2 | This study | OMR2511 | cgaagtcatggCCTCTCTTCCTC CAAGACCCCAGCTCGGTCC CGGGTGCCACCTCCTCCGC TTCTGGCCAGGGCCGCa |
| Sequence-based reagent (oligos used to clone 5'UTR of interest) | RPS6KB2 | This study | OMR2512 | agcttAGTCAGTGCGCGGC CAGGTACGGGCCGACGG GCCCGCGGGGCCGGCGC CGCCccatgactt |
| Sequence-based reagent (oligos used to clone 5'UTR of interest) | RPS6KB2 | This study | OMR2513 | cgaagtcatggGGCGGCGCCG GCCCCGCGGGGCCCGTCGG CCCGTACCTGGCCGCGCAC TGACTa |
| Sequence-based reagent (oligos used to clone 5'UTR of interest) | SP140L | This study | OMR2514 | agcttACACTGCACGCAGGCT GGGCCGACTGGGGAGCTCAT AGGCCAGGCTCTGACACCCAG GCAGGGCCTAGGGTGGGACGccatgactt |

*Continued on next page*

*Continued*

| Reagent type (species) or resource | Designation | Source or reference | Identifiers | Additional information |
|---|---|---|---|---|
| Sequence-based reagent (oligos used to clone 5'UTR of interest) | SP140L | This study | OMR2515 | cgaagtcatggCGTCCCACCCTA GGCCCTGCCTGGGTGTCAG AGCCTGGCCTATGAGCTCCC CAGTCGGCCCAGCCTGCGTG CAGTGTa |
| Sequence-based reagent (oligos used to clone 5'UTR of interest) | CLPTM1L | This study | OMR2516 | agcttGACCCGGAGCGGGAAGccatgactt |
| Sequence-based reagent (oligos used to clone 5'UTR of interest) | CLPTM1L | This study | OMR2517 | cgaagtcatggCTTCCCGCTCCGGGTCa |
| Sequence-based reagent (oligos used to clone 5'UTR of interest) | NCAPH2 | This study | OMR2518 | TAGTGAACCGTCAGATCACT AGAAGCTTGCATTTTCCTGG GCGGGAACAGCAAAATGGC GCCAGAACTAGTGGCGGGC TGAGGACGCCGTACCCCTCGGA |
| Sequence-based reagent (oligos used to clone 5'UTR of interest) | NCAPH2 | This study | OMR2519 | CTTTCGAAGTCATGGGTC CGGGAGGGAACGGGCGGC AAAGGGACCGCAGGGCTGC CTTCCGAGGGGTACGGCGTC CTCAGCCCGCCACTAGTTCTGGCGCC |
| Sequence-based reagent (oligos used to clone 5'UTR of interest) | CBX4 | This study | OMR2520 | AGAAGCTTAGTTGT CTGAGCGAGCGC |
| Sequence-based reagent (oligos used to clone 5'UTR of interest) | CBX4 | This study | OMR2521 | ACTTTCGAAGTCATGGG GCCGAGCCGGAGCG |
| Sequence-based reagent (oligos used to clone 5'UTR of interest) | TEAD3 | This study | OMR2522 | AGAAGCTTAACACAA ACTTTCCGTCCCGCTC |
| Sequence-based reagent (oligos used to clone 5'UTR of interest) | TEAD3 | This study | OMR2523 | ACTTTCGAAGTCATGGT GTGCTGGTTGCTCTGGGC |
| Sequence-based reagent (oligos used to clone 5'UTR of interest) | SLC7A11 | This study | OMR2526 | TAGAAGCttGGTTTGT AATGATAGGGCGGCAG |
| Sequence-based reagent (oligos used to clone 5'UTR of interest) | SLC7A11 | This study | OMR2527 | ccggTTCGAAgtCATggtggA GTAGGGACACACGGGGG |
| Sequence-based reagent (oligos used to clone 5'UTR of interest) | QTRT2 | This study | OMR2528 | TAGAAGCttAGTACTC CCTGATTGGCTCTGC |
| Sequence-based reagent (oligos used to clone 5'UTR of interest) | QTRT2 | This study | OMR2529 | ccggTTCGAAgtCATggtgg CCTAAGGGATTCTTCTA GGTCCTTTCAGC |
| Sequence-based reagent (oligos used to clone 5'UTR of interest) | POLR3E | This study | OMR2530 | TAGAAGCttACGTG TCCGCCGGAGTT |
| Sequence-based reagent (oligos used to clone 5'UTR of interest) | POLR3E | This study | OMR2531 | ccggTTCGAAgtCATggtggA CTAGAGGAGAGCCAGCCG |
| Sequence-based reagent (oligos used to clone 5'UTR of interest) | CNOT1 | This study | OMR2532 | TAGAAGCttGTAGAGA AACAAGCGGAGTTAACCGA |
| Sequence-based reagent (oligos used to clone 5'UTR of interest) | CNOT1 | This study | OMR2533 | ccggTTCGAAgtCATggtggTGC TGGTTGGGGCGGAA |
| Sequence-based reagent (oligos used to clone 5'UTR of interest) | PNPLA8 | This study | OMR2536 | TAGAAGCttAGTGTTTG TGTTGGAAGCTCAGC |
| Sequence-based reagent (oligos used to clone 5'UTR of interest) | PNPLA8 | This study | OMR2537 | ccggTTCGAAgtCATggtggAA CTTAAAAATCATTTATTTTCT ATGACATTCTCTCACTTCTTGA |
| Sequence-based reagent (oligos used to clone 5'UTR of interest) | ACTN4 | This study | OMR2538 | TAGAAGCttGAAGC AGCTGAAGCGGCG |
| Sequence-based reagent (oligos used to clone 5'UTR of interest) | ACTN4 | This study | OMR2539 | ccggTTCGAAgtCATggtgg TCCGCCGCCTCTCGC |
| Sequence-based reagent (oligos used to clone 5'UTR of interest) | SLC38A1 | This study | OMR2540 | CTAgatatccaACTGACA CGCAGCTTTGGTTAAA |
| Sequence-based reagent (oligos used to clone 5'UTR of interest) | SLC38A1 | This study | OMR2541 | ccggTTCGAAgtCATggtggG ATTAGAAAGTGTCTGTAG TTTGAAAATTAGTCCA |
| Sequence-based reagent (oligos used to clone 5'UTR of interest) | short 5' UTR | This study | OMR2609 | CGTTTAGTGAACCGTCA GATCACCACCATGACTT |

*Continued*

| Reagent type (species) or resource | Designation | Source or reference | Identifiers | Additional information |
|---|---|---|---|---|
| Sequence-based reagent (oligos used to clone 5'UTR of interest) | short 5' UTR | This study | OMR2610 | CGAAGTCATGGTGGTGATC TGACGGTTCACTAAACGAGCT |
| Sequence-based reagent (oligos used to clone 5'UTR of interest) | PIGG | This study | OMR277 | ccggaagcttGACGATAAGGCCTGGCG |
| Sequence-based reagent (oligos used to clone 5'UTR of interest) | PIGG | This study | OMR278 | ccggTTCGAAgtCATgtggCG TGGACACGCTAGGCT |
| Sequence-based reagent (oligos used to clone 5'UTR of interest) | ATG9A | This study | OMR293 | ccggaagcttGAGTGGCAGACACCCG |
| Sequence-based reagent (oligos used to clone 5'UTR of interest) | ATG9A | This study | OMR294 | ccggTTCGAAgtCATggtggCACCACCGCCCCCTG |
| Sequence-based reagent (oligos used to clone 5'UTR of interest) | IQGAP1 | This study | OMR472 | ccggaagcttGACCCCGGCAAGCC |
| Sequence-based reagent (oligos used to clone 5'UTR of interest) | IQGAP1 | This study | OMR473 | ccggTTCGAAgtCATggtggGGCGGACGAGCCC |
| Sequence-based reagent (oligos used to clone 5'UTR of interest) | PTP4A1 | This study | OMR504 | ccggaagcttGAGATTACTGCCAGGCACA |
| Sequence-based reagent (oligos used to clone 5'UTR of interest) | PTP4A1 | This study | OMR505 | ccggTTCGAAgtCATggtggGT TAATTTAGTTAAAAAACACT CAATAGGGTTATGAA |
| Sequence-based reagent (oligos used to clone 5'UTR of interest) | IFRD1 | This study | OMR508 | ccggaagcttGTTAAAA CCAGACTGCACTCC |
| Sequence-based reagent (oligos used to clone 5'UTR of interest) | IFRD1 | This study | OMR509 | ccggTTCGAAgtCATgg tggCGTGGGACGCCCGG |
| Sequence-based reagent (oligos used to clone 5'UTR of interest) | CCND3 | This study | OMR526 | ccggaagcttACCTATGCCGCGTGGG |
| Sequence-based reagent (oligos used to clone 5'UTR of interest) | CCND3 | This study | OMR527 | CGAAGCGGCcgcATTTCA CAATCATCTTTATTACAGTAGG |
| Sequence-based reagent (oligos used to clone 5'UTR of interest) | PPP1R15B | This study | OMR549 | ccggaagcttATTTTGGGCTTCGCTTCC |
| Sequence-based reagent (oligos used to clone 5'UTR of interest) | PPP1R15B | This study | OMR550 | ccggTTCGAAgtCATggtggACGGGATTCGGAGG |
| Sequence-based reagent (eIF2A Construct used for an overexpression) | N-terminally Flag-tagged eIF2A in pCDNA3 | This study | pMR007 | |
| Commercial assay or kit | Dual-Luciferase assay system | Promega | E1910 | |
| Commercial assay or kit | Cell Titer Glo | Promega | G7572 | |
| Commercial assay or kit | Next-Seq 550 system | Illumina | 20024906 | |
| Commercial assay or kit | Next-Flex small RNA v.4 kit protocol | Perkin Elmer | NOVA-5132–06 | |
| commercial assay or kit | Illumina TruSeq Stranded library preparation kit | Illumina | 20020594 | |
| Software, algorithm | algorithm to analyze ribosome profiling data | lab developed software *Teleman, 2025* | https://github.com/ aurelioteleman/Teleman-Lab | |
| Chemical compound, drug | sodium arsenite | Sigma | S7400-100G | |
| Chemical compound, drug | O-Propargyl-puromycin | Enzo Life Sciences | JBS-NU-931–05 | |
| Chemical compound, drug | tunicamycin | Sigma | 654380–10 MG | |

## Cell lines, culture conditions, and treatments

Cell lines were cultured in DMEM (Gibco 41965039) supplemented with 10% fetal bovine serum (FBS) (Sigma, S0615) and 100 U/ml Penicillin/Streptomycin (Gibco 15140122). Cell splitting was done with a quick PBS wash and treatment with Trypsin-EDTA (Gibco, 25200056). All cell lines were tested negative for mycoplasma and authenticated using SNP typing. For the indicated experiments, cells were treated either with DMSO or with 100 µM Sodium arsenite (Sigma, S7400-100G), 1 µg/ml poly (I:C) (Tocris Bioscience, 4287/10), 1 µg/ml lipopolysaccharides LPC (Sigma, L2630), or 1 µg/ml tuni-camycin (PanReac AppliChem, A2242,0005) for the indicated periods of time. For siRNA-mediated

knock-down, cells were reverse-transfected during seeding with 1.5 µl of 20 µM siRNA mix and 9 µl Lipofectamine RNAiMax reagent (Invitrogen 13778075). 72 hr post-transfection, cells were re-seeded in 96-well format to perform luciferase assays. Sequences of siRNAs used in this study are provided in *Supplementary file 1*.

## Generation of knockout cell lines and targeted CRISPR-Cas9 screen

eIF2A knock-out HeLa and HEK293T-H-Kb cell lines were generated using CRISPR-Cas9 with sgRNA sequences designed using CHOPCHOP software (*Labun et al., 2019*), and listed in *Supplementary file 2*. Oligos coding for the sgRNA sequences were cloned into pX459V2.0 (*Doench et al., 2016*) via the Bbs1 site, then wild-type HeLa or HEK293T-H-Kb cells were transfected with the plasmids using Lipofectamine 2000 in a ratio of 2:1 reagent:DNA (Life Technologies, 11668500). 24 hr post-transfection, transfected cells were selected with medium containing 1.5 µg/ml puromycin (Sigma-Aldrich, P9620). After 3 d, surviving cells were shifted into normal medium (1 x DMEM, 10% fetal bovine serum, 1% Penicillin/Streptomycin) and regrown to confluence. Single clones were selected by serial dilution into 96-well plates. Loss of protein in expanded single clones was tested with anti-eIF2A antibodies by immunoblotting and clones were confirmed by genotyping.

## Preparation of cell lysates with RIPA buffer

Cells were seeded at a density of 500,000 cells per six-well. Following a treatment, cells were washed briefly with PBS, and then lysed with 120 µl of RIPA buffer supplemented with 20 U of Benzonase (Merk Millipore, 70746–3), protease (Sigma, 4693159001), and phosphatase (Sigma, 4906837001) inhibitors. Cells were collected by scraping and the lysate was clarified by centrifugation at 4 °C for 10 min at 20,000 g. The protein concentration was measured by Pierce BCA (Life Technologies, 23224, 23228). Samples were balanced to equal protein concentration and mixed with 5 x Laemmli buffer (1/5 of the final volume). Samples were incubated at 95 °C for 5 min and then loaded on an SDS-PAGE gel.

## Western blotting

Cell lysates were separated on SDS-PAGE gels, and transferred to a nitrocellulose membrane with 0.4 µm pore size (Amersham, 10600002) by wet transfer. To block unspecific binding, membranes were blocked in 5% skim milk/PBST for 1 hr. Membranes were then probed by overnight incubation in primary antibody solution (5% BSA/PBST) at 4 °C. On the following day, membranes were washed three times 15' in PBST and incubated in secondary antibody (1:10,000 in 5% skim milk/PBST) for 2 hr at room temperature. To remove unbound secondary antibodies, membranes were washed three times for 15 min in PBST. Finally, chemiluminescence was detected with ECL reagents (Thermo Scientific, 32109) and imaged with a Biorad ChemiDoc imaging system. Antibodies used for immunoblotting are listed in *Supplementary file 3*.

## Cloning

Firefly (pAT1620) and Renilla luciferase (pAT1618) under control of LMNB1 5'UTR were described previously (*Schleich et al., 2017*). The 5'UTRs of eIF2A-dependent candidate transcripts, as well as integrated stress-responsive 5'UTRs, were PCR amplified from HeLa cDNA with the oligos indicated in *Supplementary file 4*. PCR products were gel purified, digested with HindIII and Bsp119I, and subsequently cloned into pAT1618 via the same sites. If the 5' UTR length was below 120 nucleotides, it was directly oligo-cloned into pAT1618 via HindIII and Bsp119i. The cloned 5' UTR variants are listed in *Supplementary file 5* with the indicated transcript ID. Reporters with uORFs with different Kozak strengths were generated and described previously (*Bohlen et al., 2023*). The reporters with uORFs starting with near-cognate start codons were generated for this study by substituting the AUG-uORF generated in *Bohlen et al., 2023* with the near-cognate one via oligo-cloning using the Kpn2I and BshT1 sites with oligos listed in *Supplementary file 4*. The EMCV-IRES reporter was generated and described previously in *Roiuk et al., 2024*. The short 5'-UTR reporter was produced by oligo cloning via SacI and Bsp119I sites in pAT1618. All generated constructs were verified by Sanger sequencing. All plasmids are available at the European Plasmid Repository.

## Ribosome profiling

Control or eIF2A[KO] HeLa cells were seeded at 1.2 million cells per 15 cm dish in 20 ml of growth medium 2 d before harvesting. The following day, cells were treated either with DMSO or 1 µg/ml tunicamycin for 16 hr. After treatment, cells were quickly washed with ice-cold 1x PBS supplemented with 10 mM MgCl2 and 800 µM Cycloheximide. After the wash, all residual solution was removed by gentle taping of the 15 cm dish on its side, followed by cell lysis with 150 µl of the following buffer: 0,25 M HEPES pH 7.5, 50 mM MgCl2, 1 M KCl, 5% NP40, 1000 µM Cycloheximide. Cells were scraped into an Eppendorf tube and the lysate was clarified by centrifugation at 15,000×g for 10 min at 4 °C. The concentration of lysate was estimated using a nanodrop spectrophotometer, measuring RNA content against a water-blanked control. 150 µl of lysate was used for the total RNA preparation with the RNeasy kit (Qiagen, cat. No. 74106). The remaining lysate was used for treatment with RNase I (100 Units per 120 µg of lysate) on ice for 30 min. Following digestion, the lysates were loaded on a 15–65% sucrose gradient, which was prepared in advance with the use of a Biocomp Gradient Master. The lysate was ultracentrifuged for 3 hr at 35,000 rpm in a Beckman Ultracentrifuge with a SW40Ti rotor. To collect the 80 S fraction the gradient was separated on a Biocomp Gradient Profiler system. The collected 80 S fractions were used for RNA extraction with acid-phenol. Briefly, the volume of the sample was adjusted to 700 µl with 10 mM Tris pH 7.0 and mixed with 750 µl of prewarmed acid phenol. Sample-phenol mix was incubated at 65° C with constant shaking at 1400 rpm for 15 min, followed by incubation on ice for 5 min. The sample was subsequently centrifuged at 20,000 g for 2 min and the supernatant was transferred into a new tube, containing 700 µl of acid phenol. After 5 min of incubation at room temperature, the sample was spun at 20,000 g for 2 min and the supernatant was transferred into the tube with 600 µl of chloroform. The sample was then mixed by vortexing and centrifuged at 20,000 g for 2 min. The supernatant fraction was transferred into a new tube and mixed with an equal volume of isopropanol, 2 µl of Glycoblue (Invitrogen AM9516), and 1/10 vol 3 M NaAc pH 5.2. To precipitate RNA, the sample was incubated overnight at –80° C. The following day, the sample was centrifuged at 4° C for 30 min at 14,000 rpm, followed by a 70% ethanol wash. The RNA pellet was resuspended in RNase- DNase-free water. The integrity of all RNA samples was analyzed on a Bioanalyser. To size-select footprints, RNA extracted from the 80 S peak was run on a 15% Urea-Polyacrylamide gel, and fragments of 25–35 nucleotides were purified from the gel. For this, the gel pieces containing the footprints were broken into small pieces with gel smasher tubes. 0.5 ml of 10 mM Tris pH 7 were added to the smashed gel pieces and the suspension was incubated at 70 °C for 10 min with shaking. The mix was briefly centrifuged and the supernatant was used for RNA precipitation by isopropanol. Purified footprints were phosphorylated by means of T4 PNK (NEB) for 1 hr at 37 °C in PNK buffer supplemented with 10 mM ATP. After this, the footprints were again precipitated and purified using isopropanol. To estimate the quality of footprints, RNA was run on an Agilent Bioanalyzer small RNA chip, followed by library preparation with the Next-Flex small RNA v.4 kit protocol (Perkin Elmer, NOVA-5132–06), in accordance with manufacturer recommendations. Total RNA libraries were prepared using the Illumina TruSeq Stranded library preparation kit. The quality of libraries were checked on a Bioanalyser with the use of a High sensitivity DNA kit (Agilent, 5067–4626). The libraries were sequenced on an Illumina Next-Seq 550 system.

## Data analyses of ribosome profiling

Reads were trimmed from adaptors and randomized nucleotides derived from use of the Nextflex kit with cutadapt software. By use of Bowtie2, the reads aligning to tRNA or rRNA were removed. All remaining reads were mapped to the human transcriptome (Ensemble transcript assembly 94) and genome (hg38) using BBMap software, with multiple mapping allowed. The reads mapping to the coding sequences were quantified with lab-based software written in C (available at https://github.com/aurelioteleman/Teleman-Lab copy archived at *Teleman, 2025*). For each transcript, the value of reads per kilobase of coding sequence was estimated and only transcripts with values more or equal to 2.5 were used for the subsequent analysis. Metagene profiles were built with the custom-made software written in C (available at https://github.com/aurelioteleman/Teleman-Lab copy archived at *Teleman, 2025*). For the metagene profile of uORF stop codons, only transcripts with a space of more than 50 nucleotides between uORF and main ORF were used. The DESeq2 software package was used to calculate log2 fold-changes and adjusted p-values for the difference in translation efficiency

(defined as ribosome footprints/total RNA) in control versus eIF2A knockout cells. DESeq2 was run with the design = ~assay + condition + assay:condition.

## OPP incorporation assay

A total of 0.5 million control or eIF2A[KO] HeLa cells were seeded in six-well plates a day prior to treatment. On the next day, cells were labeled by incubation with 20 µM OPP reagent (Jena Bioscience NU-931–05) for 30 min. Negative control sample was pre-treated with 100 µg/ml cycloheximide, and the cycloheximide was maintained during OPP labeling. Following labeling, the cells were lysed, the concentration of samples was measured and equalized, and the incorporation of OPP was estimated via western blot using anti-puromycin antibodies.

## Dual-luciferase translation reporter assay

Cells were seeded in 96-well plate format at a density of 8000 cells per well. The following day, cells were transfected with Lipofectamine 2000 with 100 ng of Renilla luciferase plasmid and 100 ng of Firefly luciferase plasmid per well. In case of tunicamycin (TM) treatment, 6 hr post-transfection, medium was exchanged with fresh medium supplemented either with DMSO or 1 µg/ml TM. After 16 hr, the luciferase assay was carried out using the Promega Dual-Luciferase assay system (Promega, E1910) following the manufacturer's instructions.

## Immunofluorescent detection of overexpressed Flag-eIF2A

15,000 control HeLa cells were seeded in a 12-well format on poly-lysine coated 12 mm coverslips. The following day, cells were transfected with a plasmid coding for eIF2A-Flag using Lipofectamine 2000 in a ratio of 1:2 DNA:reagent (Life Technologies, 11668500). One day later, cells were treated with or without sodium arsenite (100 µM, 1 hr) and then washed with PBS and fixed by incubation for 15 min in 4% formaldehyde in PBS. Cells were permeabilized by incubation for 10 min in 0.2% Triton-X-100 in PBS and blocked for 1 hr in 0.25% BSA in PBS. Following blocking, cells were incubated overnight at 4 C in primary antibodies. On the following day, the cells were washed twice with blocking buffer and incubated with fluorophore-conjugated secondary antibodies for 1 hr. Finally, cells were shortly stained with 5 µg DAPI (Applichem A1001) and mounted on glass slides using Vectashield (Vector Labs H-1000). The distribution of Flag-eIF2A was analyzed on a confocal microscope (Leika TCS SP8) using a 63 x objective.

## Cell proliferation assay

For cell proliferation assays, cells were seeded into 96 well plates at a density of 1000 cells per well in 100 µl of medium, eight wells per condition. A total of five plates were seeded for the sequential sample collection. The proliferation curve was built by harvesting one plate every 24 hr and performing the Cell Titer Glo (Promega, G7572) according to the manufacturer's instructions.

## Subcellular fractionation

On the day prior to the experiment, cells were seeded in 10 cm dish format, 1.5–2 million cells per dish. The next day, cells were treated for 2 hr either with 1 µg/ml Tunicamycin or DMSO, followed by a quick wash with PBS and harvesting by trypsinization. The collected cells were pelleted by gentle centrifugation at 4° C (3000 g 5 min). The supernatant was removed, and the cell pellet was quickly washed twice with PBS. Cells were resuspended with 200 µl of 1 x Hypotonic Buffer (20 mM Tric-HCl, pH 7.4; 10 mM NaCl, 3 mM MgCl) by gently pipetting up and down. The resuspended cells were incubated on ice for 15 min, followed by the addition of 1/20 of the suspension volume of 10% NP40. Cells with NP40 were vortexed for 10 s at the highest speed and centrifuged at 3000 g for 10 min at 4 C. The supernatant was moved into a new tube and marked as the cytosolic fraction, while the pellet was washed twice with PBS, moved into a new tube and resuspended in 200 µl of Cell Extraction Buffer (10 mM Tris, pH 7.4, 2 mM $Na_3VO_4$, 100 mM NaCL, 1% Triton X-100, 1 mM EDTA, 10% glycerol, 1 mM EGTA, 0.1% SDS, 1 mM NaF, 0.5% deoxycholate, 20 mM $Na_4P_2O_7$). The pellet was incubated on ice for 30 min with occasional vortexing, followed by centrifugation at 14,000 g at 4 C for 30 min. The supernatant was collected and marked as the nuclear fraction.

## Simultaneous detection of a uORF/oORF peptide and mainORF mNeonGreen

The HEK293T-K$^B$ *eIF2A*$^{KO}$ cell line was generated the same way as HeLa-*eIF2A*$^{KO}$ cells. Two days prior to the experiment, cells were seeded in six-well format. The following day, cells were transfected with empty vector, or with plasmids encoding mNeonGreen-PEST carrying either an oORF-less 5'UTR, or a 5'UTR containing a uORF coding for 'SIINFEKL' with either AUG or near-cognate start codons. The AUG codon was placed in a Kozak context predicted to be either 'medium' or 'strong.' 24 hr post-transfection, cells were washed with PBS, collected by trypsinization, and resuspended and incubated for 10 min in blocking buffer (1% BSA in 1x PBS). After blocking, cells were washed once more with PBS and stained for 30 min in the dark at 4° C with 0.25 µg/ml monoclonal Antibody OVA257-264, which detects the SIINFEKL-peptide bound to H-2Kb (Life technologies 25-5743-82). Following staining, cells were washed three times with blocking buffer to remove unbound antibodies, resuspended with 300 µl of blocking buffer, and analyzed with a Guava easyCyteTM flow cytometer running Guava Soft 3.3.

## Quantitative RT-PCR

Total RNA from either control of eIF2A$^{KO}$ HeLa cells were extracted using RNase Mini spin columns (Qiagen, cat. no. 74106). Reverse transcription (RT) of 1 µg of total RNA with random hexamer and oligo-dT +primers using Maxima H minus reverse transcriptase was performed to generate cDNA. The amplification efficiency of all Q-RT-PCR primer pairs was checked using serial dilution of a sample. Quantitative RT-PCR was run on a QuantStudio3 instrument with primaQUANT SYBRGreen low ROX master mix. RNA levels were normalized to the levels of either *GAPDH* or *RPL13A* mRNA. Sequences of oligos used for Q-RT-PCR are provided in *Supplementary file 6*.

## Acknowledgements

We thank the DKFZ Genomics Core Facility for next-generation DNA sequencing.

## Additional information

### Funding

No external funding was received for this work.

### Author contributions

Mykola Roiuk, Conceptualization, Resources, Data curation, Formal analysis, Validation, Investigation, Visualization, Writing – original draft, Project administration, Writing – review and editing; Marilena Neff, Formal analysis, Investigation; Aurelio A Teleman, Conceptualization, Software, Formal analysis, Supervision, Funding acquisition, Visualization, Writing – original draft, Project administration, Writing – review and editing

### Author ORCIDs

Mykola Roiuk http://orcid.org/0000-0001-5225-4422
Aurelio A Teleman https://orcid.org/0000-0002-4237-9368

Reviewer #1 (Public review): https://doi.org/10.7554/eLife.105311.3.sa1
Reviewer #2 (Public review): https://doi.org/10.7554/eLife.105311.3.sa2
Reviewer #3 (Public review): https://doi.org/10.7554/eLife.105311.3.sa3
Author response https://doi.org/10.7554/eLife.105311.3.sa4

## Additional files

### Supplementary files

Supplementary file 1. Sequences of siRNA against eIF2A used in this study. (Equimolar mix was

used.).

Supplementary file 2. Sequences of oligos used for sgRNA cloning.

Supplementary file 3. Antibodies used in this study.

Supplementary file 4. Sequences of oligos used for cloning in this study.

Supplementary file 5. Transcript IDs from which 5'UTRs were cloned in this study.

Supplementary file 6. Sequences of Q-RT-PCR oligos used in this study.

MDAR checklist

Source data 1. Source data underlying all graphs in the paper.

## Data availability

Sequencing data have been deposited at NCBI GEO with accession number GSE282509. All other data are contained in the source data files.

The following dataset was generated:

| Author(s) | Year | Dataset title | Dataset URL | Database and Identifier |
|---|---|---|---|---|
| Roiuk M, Neff M, Teleman AA | 2025 | Human eIF2A has a minimal role in translation initiation and in uORFmediated translational control | https://www.ncbi.nlm.nih.gov/geo/query/acc.cgi?acc=GSE282509 | NCBI Gene Expression Omnibus, GSE282509 |

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
