## [Editor Report · eLife Assessment]

In this **valuable** study, Roiuk et al combined ribosome profiling and reporter assays to provide **compelling** evidence that eIF2A does not have a major impact on mRNA translation in HeLa cells. These findings are consistent with several recent publications that disaffirm the previously proposed role of eIF2A in directing protein synthesis under stress. Considering that stress-dependent perturbations in translation play a major role in homeostasis and several pathological states (e.g., cancer and neurological disorders), this work should be of broad interest to researchers studying regulation of gene expression, stress-adaptation, cancer and neurobiology.

---

## [Referee Report · Reviewer #1 (Public review)]

Summary:

Beyond what is stated in the title of this paper, not much needs to be summarized. eIF2A in HeLa cells promotes translation initiation of neither the main ORFs nor short uORFs under any of the conditions tested.

Strengths:

Very comprehensive, in fact, given the huge amount of purely negative data, an admirably comprehensive and well-executed analysis of the factor of interest.

Weaknesses:

The study is limited to the HeLa cell line, which is now addressed and clearly stated by the authors.

---

## [Referee Report · Reviewer #2 (Public review)]

Summary

Roiuk et al describe a work in which they have investigated the role of eIF2A in translation initiation in mammals without much success. Thus, the manuscript focuses on negative results. Further, the results, while original, are generally not novel, but confirmatory, since related claims have been made before independently in different systems with Haikwad et al study recently published in eLife being the most relevant.

Despite this, we find this work highly important. This is because of a massive wealth of unreliable information and speculations regarding eIF2A role in translation arising from series of artifacts that began at the moment of eIF2A discovery. This, in combination with its misfortunate naming (eIF2A is often mixed up with alpha subunit of eIF2, eIF2S1) has generated a widespread confusion among researchers who are not experts in eukaryotic translation initiation. Given this, it is not only justifiable but critical to make independent efforts to clear up this confusion and I very much appreciate the authors' efforts in this regard.

Strengths

The experimental investigation described in this manuscript is thorough, appropriate and convincing.

Weaknesses

No major weaknesses as the authors have improved their presentation.

---

## [Referee Report · Reviewer #3 (Public review)]

Summary:

This is a valuable study providing solid evidence that the putative non-canonical initiation factor eIF2A has little or no role in the translation of any expressed mRNAs in cultured human (primarily HeLa) cells. Previous studies have implicated eIF2A in GTP-independent recruitment of initiator tRNA to the small (40S) ribosomal subunit, a function analogous to canonical initiation factor eIF2, and in supporting initiation on mRNAs that do not require scanning to select the AUG codon or that contain near-cognate start codons, especially upstream ORFs with non-AUG start codons, and may use the cognate elongator tRNA for initiation. Moreover, the detected functions for eIF2A were limited to, or enhanced by, stress conditions where canonical eIF2 is phosphorylated and inactivated, suggesting that eIF2A provides a back-up function for eIF2 in such stress conditions. CRISPR gene editing was used to construct two different knock-out cell lines that were compared to the parental cell line in a large battery of assays for bulk or gene-specific translation in both unstressed conditions and when cells were treated with inhibitors that induce eIF2 phosphorylation. None of these assays identified any effects of eIF2A KO on translation in unstressed or stressed cells, indicating little or no role for eIF2A as a back-up to eIF2 and in translation initiation at near-cognate start codons, in these cultured cells.

The study is very thorough and generally well executed, examining bulk translation by puromycin labeling and polysome analysis and translational efficiencies of all expressed mRNAs by ribosome profiling, with extensive utilization of reporters equipped with the 5'UTRs of many different native transcripts to follow up on the limited number of genes whose transcripts showed significant differences in translational efficiencies (TEs) in the profiling experiments. They also looked for differences in translation of uORFs in the profiling data and examined reporters of uORF-containing mRNAs known to be translationally regulated by their uORFs in response to stress, going so far as to monitor peptide production from a uORF itself. The high precision and reproducibility of the replicate measurements instil strong confidence that the myriad of negative results they obtained reflects the lack of eIF2A function in these cells rather than data that would be too noisy to detect small effects on the eIF2A mutations. They also tested and found no evidence for a recent claim that eIF2A localizes to the cytoplasm in stress and exerts a global inhibition of translation. Given the numerous papers that have been published reporting functions of eIF2A in specific and general translational control, this study is important in providing abundant, high-quality data to the contrary, at least in these cultured cells.

Strengths:

The paper employed two CRISPR knock-out cell lines and subjected them to a combination of high-quality ribosome profiling experiments, interrogating both main coding sequences and uORFs throughout the translatome, which was complemented by extensive reporter analysis, and cell imaging in cells both unstressed and subjected to conditions of eIF2 phosphorylation, all in an effort to test previous conclusions about eIF2A functioning as an alternative to eIF2.

Weaknesses:

No major issues were observed as the authors have provided additional evidence of the extent of ISR induction by tunicamycin. The discussion was also expanded to address concerns stemming from the previous version of the manuscript.

[Editors note: Reviewers and editors concluded that the authors revised the article in a satisfactory manner and no further concerns were raised]

---

## [Author Response]

The following is the authors’ response to the original reviews

**Public Reviews:**

**Reviewer #1 (Public review):**
Summary:Beyond what is stated in the title of this paper, not much needs to be summarized. eIF2A in HeLa cells promotes translation initiation of neither the main ORFs nor short uORFs under any of the conditions tested.Strengths:Very comprehensive, in fact, given the huge amount of purely negative data, an admirably comprehensive and well-executed analysis of the factor of interest.Weaknesses:The study is limited to the HeLa cell line, focusing primarily on KO of eIF2A and neglecting the opposite scenario, higher eIF2A expression which could potentially result in an increase in non-canonical initiation events.

We thank the reviewer for the positive evaluation. As suggested by the reviewer in the detailed recommendations, we will clarify in the title, abstract and text that our conclusions are limited to HeLa cells. Furthermore, as suggested we will test the effect of eIF2A overexpression on the luciferase reporter constructs, and will upload a revised manuscript.

**Reviewer #2 (Public review):**
SummaryRoiuk et al describe a work in which they have investigated the role of eIF2A in translation initiation in mammals without much success. Thus, the manuscript focuses on negative results. Further, the results, while original, are generally not novel, but confirmatory, since related claims have been made before independently in different systems with Haikwad et al study recently published in eLife being the most relevant.Despite this, we find this work highly important. This is because of a massive wealth of unreliable information and speculations regarding eIF2A role in translation arising from series of artifacts that began at the moment of eIF2A discovery. This, in combination with its misfortunate naming (eIF2A is often mixed up with alpha subunit of eIF2, eIF2S1) has generated a widespread confusion among researchers who are not experts in eukaryotic translation initiation. Given this, it is not only justifiable but critical to make independent efforts to clear up this confusion and I very much appreciate the authors' efforts in this regard.StrengthsThe experimental investigation described in this manuscript is thorough, appropriate and convincing.WeaknessesHowever, we are not entirely satisfied with the presentation of this work which we think should be improved.

We thank the reviewer for the positive evaluation. We will revise the manuscript according to the reviewer's suggestions made in the detailed recommendations.

**Reviewer #3 (Public review):**
Summary:This is a valuable study providing solid evidence that the putative non-canonical initiation factor eIF2A has little or no role in the translation of any expressed mRNAs in cultured human (primarily HeLa) cells. Previous studies have implicated eIF2A in GTP-independent recruitment of initiator tRNA to the small (40S) ribosomal subunit, a function analogous to canonical initiation factor eIF2, and in supporting initiation on mRNAs that do not require scanning to select the AUG codon or that contain near-cognate start codons, especially upstream ORFs with non-AUG start codons, and may use the cognate elongator tRNA for initiation. Moreover, the detected functions for eIF2A were limited to, or enhanced by, stress conditions where canonical eIF2 is phosphorylated and inactivated, suggesting that eIF2A provides a back-up function for eIF2 in such stress conditions. CRISPR gene editing was used to construct two different knockout cell lines that were compared to the parental cell line in a large battery of assays for bulk or gene-specific translation in both unstressed conditions and when cells were treated with inhibitors that induce eIF2 phosphorylation. None of these assays identified any effects of eIF2A KO on translation in unstressed or stressed cells, indicating little or no role for eIF2A as a back-up to eIF2 and in translation initiation at near-cognate start codons, in these cultured cells.The study is very thorough and generally well executed, examining bulk translation by puromycin labeling and polysome analysis and translational efficiencies of all expressed mRNAs by ribosome profiling, with extensive utilization of reporters equipped with the 5'UTRs of many different native transcripts to follow up on the limited number of genes whose transcripts showed significant differences in translational efficiencies (TEs) in the profiling experiments. They also looked for differences in translation of uORFs in the profiling data and examined reporters of uORF-containing mRNAs known to be translationally regulated by their uORFs in response to stress, going so far as to monitor peptide production from a uORF itself. The high precision and reproducibility of the replicate measurements instil strong confidence that the myriad of negative results they obtained reflects the lack of eIF2A function in these cells rather than data that would be too noisy to detect small effects on the eIF2A mutations. They also tested and found no evidence for a recent claim that eIF2A localizes to the cytoplasm in stress and exerts a global inhibition of translation. Given the numerous papers that have been published reporting functions of eIF2A in specific and general translational control, this study is important in providing abundant, high-quality data to the contrary, at least in these cultured cells.Strengths:The paper employed two CRISPR knock-out cell lines and subjected them to a combination of high-quality ribosome profiling experiments, interrogating both main coding sequences and uORFs throughout the translatome, which was complemented by extensive reporter analysis, and cell imaging in cells both unstressed and subjected to conditions of eIF2 phosphorylation, all in an effort to test previous conclusions about eIF2A functioning as an alternative to eIF2.Weaknesses:There is some question about whether their induction of eIF2 phosphorylation using tunicamycin was extensive enough to state forcefully that eIF2A has little or no role in the translatome when eIF2 function is strongly impaired. Also, similar conclusions regarding the minimal role of eIF2A were reached previously for a different human cell line from a study that also enlisted ribosome profiling under conditions of extensive eIF2 phosphorylation; although that study lacked the extensive use of reporters to confirm or refute the identification by ribosome profiling of a small group of mRNAs regulated by eIF2A during stress.

We thank the reviewer for the positive evaluation. We will revise the manuscript according to the recommendations made in the detailed recommendations. Regarding the two points mentioned here:

(1) The reason eIF2alpha phosphorylation does not increase appreciably is because unfortunately the antibody is very poor. The fact that the Integrated Stress Response (ISR) is induced by our treatment can be seen, for instance, by the fact that ATF4 protein levels increase strongly (in the very same samples where eIF2alpha phosphorylation does not increase much, in Suppl. Fig. 5E). We will strengthen the conclusion that the ISR is indeed activated with additional experiments/data as suggested by the reviewer.

(2) We agree that our results are in line with results from the previous study mentioned by the reviewer, so we will revise the manuscript to mention this other study more extensively in the discussion.

**Recommendations for the authors:**

**Reviewer #1 (Recommendations for the authors):**
(1) I suggest to state (already in the abstract, but perhaps also even in the title, definitely in the rest of the paper) that this analysis is limited to the HeLa cell line.

As suggested, we have now specified in both the title and the abstract that the work is done in HeLa cells.

(2) In my view, it is a pity that the authors - given the tools are available - did not check the impact of high eIF2A levels on expression of individual mRNAs under normal and stress conditions. I am not suggesting to repeat ribo-seq in this setup, it would be too much to ask for, but re-examining some of the many reporters the authors generated with eIF2A overexpressed may point to some function, e.g. increased number of non-canonical initiation events (non-AUG-initiated)? If anything, the use of HeLa and the primary focus on eIF2A KO neglecting the prospective impact of eIF2A overexpression should be mentioned as two main limitations of this study.

We thank the reviewer for the good suggestion to test our synthetic reporters with eIF2A overexpression. New Suppl. Fig. 4G now shows that overexpression of eIF2A does not affect translation of synthetic reporters carrying an ATG start codon in different initiation contexts, or carrying near-cognate start codons, in agreement with a lack of effect on translation which we previously observed with loss of eIF2A.

(3) Ribo-seq with eIF2A. Did the authors focus on ORFs that are known, or whose isoforms are known, to be non-AUG initiated? Would the loss of eIF2A decrease FPs in their CDSes under at least some conditions?

We have now assessed the read distribution on the eIF4G2 transcript in both the control and tunicamycin conditions (Author response image 1). In our hands, eIF4G2 is one of the best examples of non-AUG initiation in human cells, since the main coding sequence starts with GTG and the CDS is well translated. Nonetheless, we do not observe any significant changes in read distribution (panels A-B) or overall translation efficiency of eIF4G2 upon eIF2A loss (panels C-D).

**Author response image 1. sa4fig1:** (A-B) Average reads occupancy on the eIF4G2 (ENST0000339995) transcript in DMSO treated (panel A, n=3) or tunicamycin treated samples (panel B, n=2) derived from either control (black) or eIF2A-KO (red) HeLa cells. Reads counts were normalized to sequencing depth and averaged between either 3 (DMSO-treated) or 2 (tunicamycin-treated) replicates. Graphs were then smoothened with a sliding window of 3 nt. (C-D) The total number of reads mapping to the eIF4G2 CDS, normalized to library sequencing depth per replica was quantified. No significant difference between control and eIF2A-KO cells was observed in either DMSO treated (panel C) or tunicamycin treated (panel D) cells. Significance by unpaired, two-sided, t-test. ns = not significant.

Thank you for giving me the opportunity to review this article.

**Reviewer #2 (Recommendations for the authors):**
While some of our suggestions below may be considered subtle, in our opinion they are important and it would be good if the authors consider them for their revision, we also have a couple of technical suggestions.(1) Abstract.The authors failed to identify the role of eIF2A in translation initiation and have provided compelling evidence that eIF2A is not involved in recognition of non-AUG codons as start codons nor in recruitment of initiator tRNA during stress conditions which are two activities most commonly misattributed to eIF2A. However, they have not exhausted all possible potential functions of eIF2A, see below, it is also possible that eIF2A may have a role not yet suggested by anyone and it may function in translation initiation in special circumstances that have not been tested yet. The authors indeed discuss such possibility in the Discussion section. Given that there is genetic evidence (that is unaffected by biochemical impurities) linking eIF2A to other initiation factors (5B and 4E), we are not yet convinced that eIF2A does not have any role in translation initiation and therefore we find the last sentence of the abstract premature. We suggest to soften this statement into something like this: whether eIF2A has any role in translation remains unknown, it may even have a role in a different aspect of RNA Biology.

We agree with the reviewer. We changed the last sentence of the abstract to read as follows:

“It is possible that eIF2A plays a role in translation regulation in specific conditions that we have not tested here, or that it plays a role in a different aspect of RNA biology.”

(2) Recently eIF2A has been implicated in ribosomal frameshifting, see Wei et al 2023 DOI: 10.1016/j.celrep.2023.112987Could authors look into PEG10 mRNA ribosome profile to see if there are detectable statistically significant changes in footprint density downstream of frameshift site between WT and eIF2A Kos? It is likely that the coverage will be insufficient to give a definitive answer, but it is worth checking, it would be a pity to miss it.

We thank the reviewer for this suggestion. We have now looked at the distribution of ribosome footprints on the PEG10 transcript variant that is expressed in HeLa cells (ENST00000482108) and indeed observe coverage downstream of the annotated stop codon, consistent with a frameshifting event that results in an extended protein isoform being translated. Visual assessment of the read distribution between the main ORF and the "ORF extension" does not show a substantial difference between control and eIF2A knock-out cells (Author response image 2A-B). Additionally, we quantified the ratio of reads mapping to the PEG10 ORF upstream of the slippery site versus those mapping downstream, extending into the predicted longer protein. Nonetheless, we could not detect significant changes between control and eIF2A-KO cells in either tested condition (Author response image 2C-D).

**Author response image 2. sa4fig2:** (A-B) Average reads occupancy on the PEG10 (ENST00000482108) transcript in DMSO treated (panel A, n=3) or tunicamycin treated samples (panel B, n=2) derived from either control (black) or eIF2A-KO (red) HeLa cells are shown. Reads counts were normalized to sequencing depth and averaged between either 3 (DMSO-treated) or 2 (tunicamycin-treated) replicates. Graphs were then smoothened with a sliding window of 3 nt. (C-D) The ratio of reads mapping to the ORF upstream of the slippery site to reads mapping to the predicted extended protein downstream to the slippery site is shown. Reads counts were normalized to the sequencing depth. Neither DMSO treated samples (panel C) nor tunicamycin treated samples (panel D) had a significant difference between control and eIF2A-KO cells. Significance by unpaired, two-sided, t-test. ns = not significant.

(3) IntroductionGiven the volume of unreliable claims regarding eIF2A in the literature and the overall confusion it is very difficult (may even be impossible) to write a clear coherent introduction into the topic. Nonetheless, there are few points that need to be taken into account.The authors state that eIF2A is capable to recruit initiator tRNA citing Zoll et al 2002. This activity was later shown to be a biochemical artefact (which was most likely reproduced by Kim et al 2018), eIF2A fraction was contaminated with eIF2D which does bind tRNAs in GTP-independent manner. eIF2A purified from RRL separates from initiator tRNA binding activity, see Dmitriev et al 2010 DOI: 10.1074/jbc.M110.119693. This point is also relevant to the second paragraph of Discussion, it should be acknowledged that it has been shown previously that eIF2A does not bind the initiator tRNA.

We appreciate the advice provided by the reviewer. We have modified both the introduction and the 2nd paragraph of the discussion to reflect that the tRNA-binding activity is due to contaminating eIF2D rather than eIF2A.

In many cases the authors describe certain claims as facts even though they refute them themselves. For example"Such eIF2A-driven non-AUG initiation events were shown to play a crucial role in different aspects of cell physiology and disease progression: cellular adaptation during the integrated stress response (Chen et al., 2019; Starck et al., 2016)" While non-AUG initiation events do play crucial roles in different aspects of cell physiology (reviewed in Andreev et al 2023 doi: 10.1186/s13059-022-02674-2) eIF2A has nothing to do with it as the authors show themselves. Therefore different language should be used, e.g.. "eIF2A has been suggested (or proposed or reported) to be responsible for non-AUG initiation events that were shown to play ..."The word "shown" is used in many other instances for the claims that the authors refute. "Shown" is only appropriate for strong evidence that leaves little doubt.

We agree with the reviewer and made the suggested changes in the text.

(4) Supplementary Fig. 1.Panel C is used to argue that eIF2A has a higher concentration than in the nucleus, perhaps it is worth explaining how this conclusion was drawn. If levels in cytoplasm are comparable to GAPDH and Tubulin but less than c-Myc in nucleus does it really mean that there is less eIF2A in the nucleus than in cytoplasm? This is not obvious to us. Also, presumably WCL stands for Whole Cell Lysate, it would be nice to introduce this abbreviation somewhere.

To compare levels of eIF2A in the nuclear and cytosolic fractions, we lysed the two fractions in equal volumes of buffer (i.e. the cytosolic fraction was extracted in 200 µl of hypotonic buffer, and the nuclear fraction was extracted in 200 µl of cell extraction buffer). This assures that per microliter of lysate we have the same number of "cytosols" or nuclei. Hence, equal intensity bands in the cytosolic and nuclear fractions would mean that half of the protein is in the nucleus and half is in the cytosol. We originally described this in the Methods section, but now also mention it in the Results and in the figure legend.

We replaced WCL with "whole cell" in the figure.

(5) The differential translation analysis is described very briefly "To obtain values of translation efficiency, log2 fold changes, and adjusted p values the DESeq2 software package was used". Was TE calculated based on ribosome footprint to RNA-seq ratios? How exactly DESeq2 was used here? TE measured in this way spuriously correlates with RNA-seq values, see Larsson et al 2010 DOI: 10.1073/pnas.1006821107, perhaps it would be worse assessing differential translation with anota2seq (Oertlin et al 2019 doi: 10.1093/nar/gkz223.)? Anota2seq avoids calculating the ratios and enables comprehensive analysis of differential translation including detection of buffered translation which might be the case here while avoiding artefacts that may arise from varying RNA levels.

We now specified in more detail in the Methods section how we analyzed the data. Indeed, the DeSeq2 was used on translation efficiency values, which we calculated as the ratio of ribosome footprints to RNA-seq.

As suggested, we have now also performed the analysis using anota2seq (Suppl. Fig. 3C) and this analysis identified zero transcripts that are translationally regulated, in agreement with our analysis.

(6) Section "eIF2a-inactivating stresses do not redirect tRNA delivery function to eIF2A."The description of ISR mechanism is a bit inaccurate. Strictly speaking eIF2alpha phosphorylation does not inactivate it eIF2alpha. It results in formation of a very stable eIF2*GDP*eIF2B complex, thus severely depleting eIF2B which serves as a GEF for eIF2. This in turn reduces the ternary complex (eIF2*GTP*tRNAi) concentration since there is no free eIF2B to exchange GDP for GTP. Without getting into much detail, we think it would be more accurate to say that eIF2alpha phosphorylation leads to ternary complex depletion instead of saying that stress inactivates eIF2alpha.

We agree with the reviewer - we were trying to use simple, compact wording. We have now reworded the section title to "No detectable role for eIF2A in translation when eIF2 is inhibited" and rephrased the subsequent text to be correct.

Also the subtitle uses eIF2a with small a that stands for alpha which potentially could lead to substantial confusion since in this case the difference between eIF2alpha and eIF2A is only in capitalisation of the last letter, many text-mining engines such as modern LLMs may not be able to pick the differences. Perhaps it would be better to refer to eIF2alpha by the HGNC approved name of its gene - eIF2S1 to avoid further confusions. For clarity it may be stated at the beginning that eIF2S1 is commonly known as eIF2alpha.

We thank the reviewer for this point. We have removed all instances of eIF2a (with lowercase a) from the manuscript to avoid this source of confusion. In the first instance of eIF2a we also added the official HGNC gene name. However, we prefer to use eIF2a instead of eIF2S1 because people outside the translation field tend to know the subunit as eIF2a, and we think it is important that also people outside the translation field read this manuscript, since some of the questionable papers on eIF2A come from labs working at the interface between translation and other fields.

MinorIntroduction(7) "uses the CAT anticodon" change CAT to CAU

We corrected CAT to CAU

(8) "In the canonical initiation pathway", change "canonical" to "most common", canonical is somewhat a judgemental statement that originates in theology. Same applies to numerous occurrences of "canonical AUG", simply using "AUG" would be simpler and more accurate as you will avoid giving impression that there are "non-canonical AUGs".

Done.

(9) "eIF2A was initially considered to be a functional analogue of prokaryotic IF2 (Merrick and Anderson, 1975), however later this role was reassigned to the above-mentioned heterotrimeric factor eIF2 (a,b,g) (Levin et al., 1973)." - there is a chronological contradiction within this sentence, the initial consideration is attributed to 1975 while its later reassignment to 1973.

We are grateful to the reviewer for spotting this mistake. There was a citation problem; we fixed it and now cite the correct paper for the initial discovery of eIF2A to PMID 5472357 (Shafritz et al 1970).

(10) "On the other hand, studies on the role of eIF2A on viral IRES translation have arrived at conflicting results." Remove "On the other hand" since conflicting results have been mentioned above. In fact the entire sentence is somewhat redundant given prior "For example, eIF2A has been studied in the context of internal ribosome entry sites (IRES), where it was found to act both as a suppressor and an activator of IRESmediated initiation."

We have rewritten the paragraph to make it more coherent.

(11) Fig. 1. C-D. is using CHX abbreviation for cycloheximide, this need to be mentioned on the legend or elsewhere in the text. Otherwise CHX may not be clear for a reader uninitiated in ribosome profiling.

We now mention in the figure legend that CHX stands for cycloheximide and indicate that it was used as a negative control to block translation.

(12) Page 7, section "Ribosome profiling reveals a few eIF2Adependent transcripts"In this section you describe ribosome profiling experiments and identify few transcripts whose translation seems to be changing based on ribosome profiling data. Then you attempt to verify them using gene expression reporters and reasonably suggest that these are false positives. In essence this section argues that there are no eIF2A-dependent transcripts, therefore the title of this subsection is misleading, it makes sense to rename it so that it better reflects the content of this section.

We agree and have renamed the section to "Ribosome profiling identifies no eIF2Adependent transcripts"

(13) Page 8, top. Rephrase "To do this, we performed ribosome profiling on control and eIF2AKO cells, which sequences the mRNA footprints protected by ribosomes."

Fixed.

(14) Page 10, bottom. "Several studies have reported that eIF2A can delivery alternative initiator tRNAs to uORFs with nearcognate start codons". Change "delivery" to "deliver".

Thanks for spotting it. We corrected to “deliver”

(15) Page 13 "This suggests that, as in non-stressed conditions, eIF2A has a minimal effect on global translation also when eIF2a activity is low." - rephrase to avoid impression that eIF2alpha activity is low in normal conditions, also please see comment #6 above.

We fixed this sentence to read: “This suggests that, as in non-stressed conditions, eIF2A has a minimal effect on global translation also when the integrated stress response is active.”

**Reviewer #3 (Recommendations for the authors):**
- The experimental data in Fig. S5E do not support the claim of increased eIF2 phosphorylation on TM treatment; although, comparing Fig. S5A with Fig. 1B supports a marked reduction in bulk translation and the reporter data in Fig. 4A show the expected induction of the uORF-containing reporters by TM. Because these are the conditions employed for ribosome profiling in stress conditions shown in Fig. 4B, it would be reassuring to document TM-induced translational efficiencies of ATF4 and the other known mRNAs resistant to eIF2 phosphorylation in the ribosome profiling data, including gene browser images of the replicate experiments. If the induction of TEs by TM for such mRNAs was not robust, it would be valuable to repeat the analysis using arsenite (SA) treatment, which produces a greater inhibition of bulk translation.

Unfortunately, the eIF2alpha antibody is not very good and also detects the nonphosphorylated protein, causing high background and poor apparent induction in response to tunicamycin. The fact that the ISR was activated is visible from the induction of ATF that was assessed by western blot in the Suppl. Fig. 5E. To ensure that our ribosome profiling libraries also recorded the activation of ISR we built single gene plots for ATF4 both in control and HeLa eIF2A-KO cell. As shown in Author response image 3 A&B in both cell lines tunicamycin treatment led to the induction of ATF4. This can also be seen by the 4-fold induction in ATF4 translation efficiency in response to tunicamycin in both WT and eIF2A-KO cells (Author response image 3C). Additionally, we checked that another marker induced by tunicamycin, HSPA5, is also translationally upregulated in both cell lines, as well as the downstream target of ATF4 – PPP1R15B. (Author response image 3C).

**Author response image 3. sa4fig3:** (A-B) Average read occupancy on the ATF4 (ENST00000674920) transcript in DMSO treated (n=3) or tunicamycin treated samples (n=2) derived from either control (panel A) or eIF2A-KO (panel B) HeLa cells are shown. Read counts were normalized to sequencing depth and averaged between either 3 (DMSO-treated) or 2 (tunicamycin-treated) replicates. Graphs were then smoothened with a sliding window of 3 nt. (C) Scatter plot of log2(fold change) of Translation Efficiency TM/DMSO for control cells on the xaxis versus eIF2AKO cells on the y-axis. The induction of ATF4 as well as the downstream target PPP1R15B are shown. The upregulation of HSP5A translation, the other hallmark of ER-stress induced by tunicamycin treatment is shown.

- It should be pointed out in the text that in both published studies being cited here of cells lacking eIF2A, that by Gaikwad et al. on a yeast eIF2A deletion mutant, and that by Ichihara et al. on human HEK293 CRISPR KO cells, the analyses included stress conditions in which eIF2 phosphorylation is induced (amino acid starvation or SA treatment, respectively), as was conducted here.

Good point - we added this information into the introduction:

"Furthermore, loss of eIF2A in several systems did not recapitulate these effects on non-AUG initiation in either non-stressed or stress conditions (caused either by amino acid depletion or sodium arsenate treatment) (Gaikwad et al., 2024; Ichihara et al., 2021)."

- The Ichihara et al. (2021) study just mentioned reached some of the same conclusions for HEK cells obtained here by conducting ribosome profiling in untreated and SA-treated cells, finding only 1 mRNA (untreated) or four mRNAs (SA-treated cells) that showed significantly reduced TEs in the eIF2A knockout vs. parental cells. It seems appropriate for the authors to expand their treatment of this prior work by summarizing its findings in some detail and also noting how their study goes beyond this previous one.

We have added a paragraph to the discussion pointing out that our data agree fully with Ichihara et al. (2021), and that Ichihara et al. (2021) also found only very few mRNAs that change in TE upon loss of eIF2A in either non-stressed or stressed conditions.